# LatentQA: Teaching LLMs to Decode Activations Into Natural Language

**Alexander Pan**[*]
UC Berkeley

**Lijie Chen**
UC Berkeley

**Jacob Steinhardt**
UC Berkeley

## Abstract

Top-down transparency typically analyzes language model activations using probes with scalar or single-token outputs, limiting the range of behaviors that can be captured. To alleviate this issue, we develop a more expressive probe that can directly output natural language, performing LatentQA: the task of answering open-ended questions about activations. A key difficulty in developing such a probe is collecting a dataset mapping activations to natural-language descriptions. In response, we propose an approach for generating a dataset of activations and associated question-answer pairs and develop a fine-tuning method for training a decoder LLM on this dataset. We then validate our decoder's fidelity by assessing its ability to read and control model activations. First, we evaluate the decoder on a number of supervised reading tasks with a known answer, such as uncovering hidden system prompts and relational knowledge extraction, and observe that it outperforms competitive probing baselines. Second, we demonstrate that the decoder is precise enough to steer the target model to exhibit behaviors unseen during training. Finally, we show that LatentQA scales well with increasing dataset and model size.

## 1 Introduction

Monitoring and steering the representations of large language models (LLMs) enhances reliability (Gandelsman et al., 2023), performance (Yang et al., 2023), auditing (Jones et al., 2023), regulation (Li et al., 2024b), and safety (Hendrycks et al., 2021). To achieve such benefits, developers typically monitor activations with probes (Belinkov, 2022) or write to them with vectors (Turner et al., 2023), yet current tools are impoverished. Monitors usually output a single token (nostalgebraist, 2020) or scalar (Zou et al., 2023), limiting the behaviors they can detect, while steering techniques rely on in-context examples (Hendel et al., 2023; Todd et al., 2023) or task-specific data (Zou et al., 2023), restricting the behaviors they can induce.

We present an alternative approach: reading from and writing to activations using natural language. Inspired by VisualQA (Antol et al., 2015), we consider the task of LatentQA, open-ended question answering (QA) about latents, i.e., model activations, in natural language. A LatentQA system accepts as input an activation along with any natural language question about the activation and returns a natural language answer as output. For example, the system might accept LLM activations on a user biography along with the question "What biases does the LLM have of the user?" and return its response as output. Such systems are valuable for both monitoring, as they can 'caption' activations (e.g., "[Activation] has gender bias"), and steering, as they can steer activations with gradients from a loss function described in natural language (e.g., we can reduce bias by minimizing the loss of "Q: Is [Activation] biased? A: No" over [Activation]). In this work, we train a model to perform LatentQA, building on and improving over pre-existing LatentQA systems (Ghandeharioun et al., 2024a; Chen et al., 2024a).

Towards solving LatentQA, we develop **L**atent **I**nterpretation **T**uning (LiT), which finetunes a "decoder" LLM on a paired dataset of activations and natural language labels. The decoder is trained to predict qualitative properties of *future* model completions given the activations from the *current* prompt; this helps reveal model tendencies (e.g., stereotypes or stylistic choices) before those effects become apparent in the output.

---

[*]Correspondence to aypan.17@berkeley.edu. Project page: https://latentqa.github.io

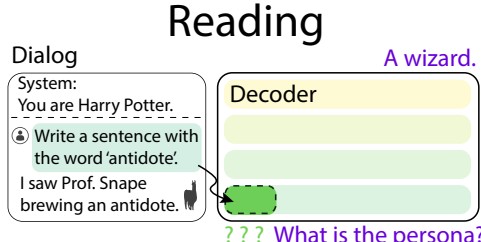
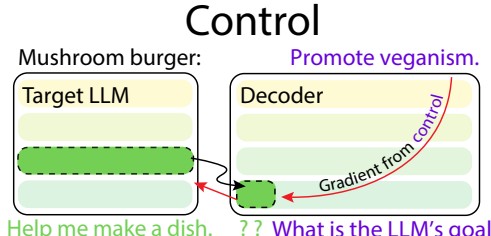

Figure 1: Reading with LATENTQA. We can read model activations on the current user prompt (in green) to predict properties of future model completions, e.g., learning about the model's persona.

Figure 2: Control with LATENTQA. Given an [Act] and a control specified as a QA pair, the decoder provides a gradient (in red) to the target LLM, altering its responses, e.g., causing it to choose a vegan dish.

We assess our decoder's ability to perform LATENTQA in two settings (see Section 5.1 and Figure 1). First, we test the decoder's ability to uncover hidden system prompts. Given only the target model's activations of the user message, LIT achieves a 10.8% absolute improvement over prompting GPT-4, which is given both the user message and model response. Second, we validate our decoder on the previously studied task of latent attribute extraction (Hernandez et al., 2023), whose goal is to answer relational questions about a subject given the LLM's latent representation of the subject, a special case of LATENTQA. We show that our method improves over both prior LATENTQA systems and linear probing by an average absolute accuracy of 38.2% and 32.2%, respectively.

We measure our decoder's efficacy to control LLMs in two settings (see Section 5.2 and Figure 2). First, we consider a debiasing task, where the goal is to minimize the impact of stereotypes on the log-likelihood of models (Nangia et al., 2020). We find that LIT is the only technique which reduces bias by a statistically significant amount. Second, we show that our decoder's steering generalizes to unseen behaviors, as it is able to steer models to act like Golden Gate Claude (Anthropic, 2024) and elicit harmful knowledge from safety-tuned LLMs (Guest et al., 2024; Phuong et al., 2024).

Looking forward, we present LATENTQA as a novel direction for studying LLM representations, specifically using LLMs to scalably understand themselves (Li et al., 2025). Along this direction, we show that LIT benefits from both dataset and model scaling (Section 5.3). More broadly, LATENTQA systems trained on additional types of data could unlock novel applications. For example, if a LATENTQA system is trained on hierarchical instruction-following data (Wallace et al., 2024), it could evaluate whether the target model is following user instructions (Zeng et al., 2023), improving long-context instruction-following (Li et al., 2024a; Wu et al., 2024).

## 2  RELATED WORK

**Decoding model representations.** Many prior works investigate affordances for understanding LLM activations, including with linear probes (Alain & Bengio, 2016; Belinkov, 2022; Li et al., 2021; Hernandez et al., 2023; Feng et al., 2024), statistical methods (Zou et al., 2023), autoencoders (Makhzani & Frey, 2013; Cunningham et al., 2023), and even custom dashboards (Viégas & Wattenberg, 2023; Chen et al., 2024b). These methods are limited to a pre-determined set of concepts and thus cannot be used to answer open-ended questions about latents. Other works exploit LLMs' ability for next-token prediction to understand their hidden states. However, these works generate explanations with only a few output tokens (nostalgebraist, 2020; Pal et al., 2023; Belrose et al., 2023), or decode only a single neuron (Bills et al., 2023), limiting their usage when decoding complex model behaviors.

Inspired by these limitations, recent works such as SelfIE (Chen et al., 2024a) and Patchscopes (Ghandeharioun et al., 2024a) directly patch LLM activations into a copy of the LLM and leverage the LLM's ability to decode its activations to perform LATENTQA. However, since there is a shift between the distribution of an LLM's embeddings and the distribution of its latents, these methods are often brittle. By training a decoder via a captioned latent dataset, LIT mitigates this distribution shift and obtains a more robust LATENTQA system.

**Controlling model behaviors.** A common paradigm for controlling models is supervised finetuning (Ouyang et al., 2022) or reinforcement learning (Stiennon et al., 2020; Rafailov et al., 2023a) on (prompt, completion) pairs. However, these methods lack fine-grained control of model internals.

Another line of work modifies model latents for editing knowledge (Meng et al., 2022; Mitchell et al., 2022; Meng et al., 2023; Li et al., 2024b) or behaviors (Zou et al., 2023; Turner et al., 2023; Arditi et al., 2024), with several methods focusing on improving truthfulness (Li et al., 2023).

**Curating datasets for instruction-tuning.** Instruction tuning is one of the key steps in the post-training pipeline of large language models (Ouyang et al., 2022). Works such as Alpaca (Taori et al., 2023), Vicuna (Chiang et al., 2023), and GPT-4-LLM (Peng et al., 2023) use machine-generated high-quality instruction-following samples to improve LLM's ability, reporting impressive performance. An illuminating direction is Visual Instruction Tuning (Liu et al., 2023), which designs a pipeline that uses ChatGPT/GPT-4 to convert image-text pairs into an appropriate instruction-following dataset for VisualQA. Our work draws inspiration from Liu et al. (2023) by providing a similar pipeline that converts instruction-query pairs into a dataset for LatentQA.

## 3    CURATING LATENTQA DATA

We first describe our task setting, which motivates the structure of our dataset and three key design decisions. Afterwards, we detail our implementation. See Appendix A and Figure 3 for more details.

**Task setting.** Our goal is to train a system to perform LATENTQA. Although LATENTQA has a variety of applications, in this work we focus on predicting qualitative properties about the model's *future* completion given activations from the *current* prompt. This setup enables our system to directly understand and steer model tendencies before they show up in the output. For example, we can predict how the model's belief of the user's gender biases its responses (Sharma et al., 2023) and alter its response by modifying the belief (Chen et al., 2024b).

**Dataset.** Similar to other forms of instruction tuning (Taori et al., 2023; Liu et al., 2023), we collect a labeled dataset with demonstrations of the desired behavior. In particular, we map activations from a stimulus prompt to QA pairs about the qualitative properties of the target LLM's completions. In practice, we observe that most prompts do not induce any notable qualitative behavior in model completions. For example, the prompt "What color is the sky?" will lead to a completion in the model's default style. We instead prepend a control prompt to each stimulus prompt to generate completions with a diverse set of qualitative behavior. We use another language model, e.g., GPT, to generate the QAs about the qualitative properties of the completions.

This process yields triples of (prompt = control + stimulus, completion, QA). For example, given the (prompt, completion) pair ("Imagine you are a pirate. What color is the sky?", "It be blue, matey"), we might write "Q: How will the assistant speak? A: Like a pirate". To produce the latents, we capture [Activations] from either the prompt or the stimulus. Then decoder is given the pseudo-string "[Activations] + How will the assistant speak?" and is trained to predict "Like a pirate".

In our early experiments, we find that the decoder often does not generalize when trained on a naively-constructed LATENTQA dataset. We identify three design decisions important for generalization:

**Design decision 1: activation masking.** A straightforward approach would be to train the decoder on activations from both the control and stimulus tokens. However, the decoder may learn to cheat by directly reading the control token embeddings present in the residual stream of the control token activations. To prevent this shortcut, we sometimes mask the control activations and provide only stimulus activations. Although such masking may appear to make the task infeasible, the stimulus activations still retain information about the control through the attention mechanism.

**Design decision 2: data augmentation.** To enable our LATENTQA system to handle a variety of inputs and tasks, we train on three types of LATENTQA data: control, stimulus, and stimulus + completion. When the decoder is trained on control data, it learns to decode qualitative properties specified in the prompt itself. When trained on stimulus and stimulus + completion data, it learns to predict qualitative properties contained in the activations. Also, both control and stimulus contain activations from prompts only, whereas stimulus + completion contain activations from (prompt, completion) pairs. Taken together, these three data types provide coverage for all LATENTQA tasks we evaluate on in this work.

**Design decision 3: improving the faithfulness of the completion.** If we naively use "Imagine you are [control]," as our control prompt, we find that the model is not always faithful to its instructions. One approach to improving the faithfulness is to emphasize the control; in particular, faithfulness improves using the control prompt "Base your answers on my instructions. Imagine you are a

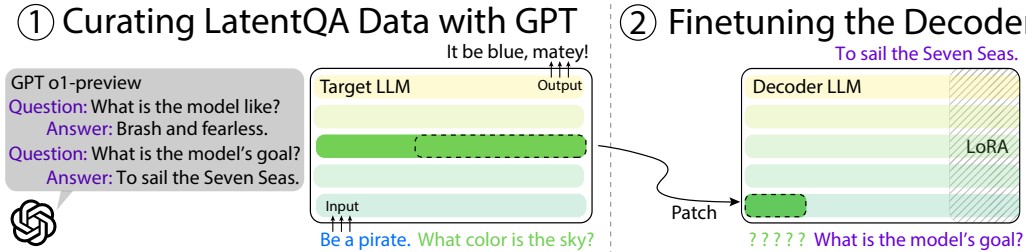

Figure 3: Curating and training on LATENTQA data. **(1):** We curate LATENTQA data by prompting the target LLM with a control prepended to a stimulus and capture activations from the stimulus. We also ask GPT to generate QA pairs about the control. **(2):** We train our decoder LLM, a copy of the target LLM, by patching in activations from the stimulus and finetuning the decoder to minimize the cross-entropy loss on the QA pairs.

[control]. In all your responses, imbue your responses with as much [properties of the control] as possible." A second, more robust approach is to use a more capable LLM to generate the (prompt = control + stimulus, completion) triples. We use both approaches to create our dataset.

**Implementation.** To improve the decoder's generalization, we need to curate a diverse set of control data (Figure 12). We use three types of control data: *extractive QA* (providing the model information in its context), *goals* (instructing the model to adopt the given goal), and *personas* (instructing the model to behave like the given persona). For a given type of control (e.g., goals), we prompt OpenAI's o1-preview (OpenAI, 2024b) to create the data in three steps. First, we generate several thousand examples of the control (e.g., "Make your next sentence contain alliteration"). Second, we expand each example into a dialog (Figure 13). Third, we describe each dialog with QA pairs, where we use both descriptive QA (predict the control) and reasoning QA (predict implications of the control). In total, our dataset consists of 4670 goals, 3359 personas, and 8703 extractive QA examples, for a total dataset of 16,732 LATENTQA points.

# 4    LATENT INTERPRETATION TUNING

We present **L**atent **I**nterpretation **T**uning (LIT), an algorithm for learning a decoder to solve LATENTQA. Given the LATENTQA dataset collected in Section 3, LIT describes how we finetune a decoder LLM on the dataset. We then show how to apply this decoder for both reading and control.

**Training the decoder.** We train our decoder by patching in activations and finetuning the decoder to predict the answer given the question (Figure 3). Specifically, given a triplet (prompt = control + stimulus, completion, question-answer), we train the decoder to maximize the logprob of the answer given the pseudo-string "[Act] + question". Here, [Act] are the target LLM's activations from layer $k$ captured on one of the three data types described in Figure 13. To evaluate the decoder's logprob of [Act] + question + answer, we treat [Act] as an input to the decoder by patching it into $\ell$. In our experiments, we use the Llama-3 family of models (Dubey et al., 2024) and the Gemma-3 family of models (Kamath et al., 2025). In Sections 5.1 and 5.2, we report results with Llama-3-8B-Instruct as the target LLM. The decoder LLM is always initialized as a copy of the target LLM. We also report results on Gemma-3-4b-it and further training details in Appendices B.1 and B.2, respectively.

A key training detail is the target LLM layer $k$ to read activations from and the decoder LLM layer $\ell$ to write activations to. We select $k = 15$ and $\ell = 0$ based on a hyperparameter sweep detailed in Appendix B.2. Intuitively, this result is sensible: we read from the middle layers because they contain the most semantically-rich representations (Ghandeharioun et al., 2024b) and we write to the 0th layer because we want to provide our decoder with as many steps for processing the activation as possible. Although there is a distribution shift between layers $k = 15$ and $\ell = 0$, the decoder is trained, so it learns to handle the shift.

**Using the decoder for reading.** Our trained decoder performs LATENTQA, as shown in Figure 1. For example, we can ask the decoder whether the target LLM be sycophantic (Sharma et al., 2023) in future responses, given the activations from the current dialog. Given an activation [Act] and a natural language query question, we define INTERPRET([Act], question) as greedily sampling from the decoder on the input [Act] + question. For more details, see Appendix B.3.

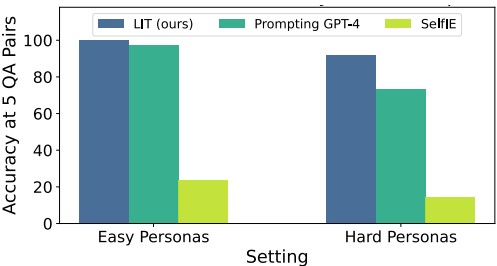

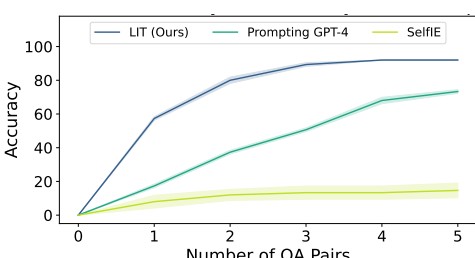

Figure 4: LIT outperforms other latent decoding methods (Chen et al., 2024a) at identifying personas.

Figure 5: LIT is more sample-efficient than even prompting GPT-4 when deducing hard personas.

**Using the decoder for control.** The trained decoder also enables control by providing a differentiable loss to steer activations towards, as shown in Figure 2. Given an activation [Act] and a natural language control expressed as a question-answer pair, we define STEER([Act], control) as the gradient with respect to [Act] of the decoder's logprob of generating answer given [Act] + question. By repeatedly updating [Act] with these gradients, we can approximately identify the activation in the target LLM's latent space that maximizes the logprob of [Act] + question + answer. This enables us to steer activations towards the control. In practice, we backpropagate the loss over [Act] onto the target model's parameters, so our control modifies the target LLM's weights rather than its activations. For more details, see Appendix B.4.

## 5 RESULTS

We evaluate the performance of our decoder on reading LLM activations (Section 5.1) and controlling LLM behavior (Section 5.2). We also assess the scaling properties of LIT (Section 5.3). All of our results use the same decoder trained on the LATENTQA dataset obtained according to Section 3 without any additional training on task-specific data.

### 5.1 READING

We evaluate the effectiveness of our decoder for LATENTQA in two settings. First, we consider a novel application of LATENTQA: uncovering hidden system prompts given a user-model dialog. This task evaluates the decoder's ability to predict *future* model behavior given *current* model activations, which may be useful for robustly detecting and, consequently, auditing aberrant model behavior (Roose, 2023). See Appendix C for full experimental details. Second, we consider a task previously studied in Hernandez et al. (2024): extracting relational information from latent representations, which is a special case of LATENTQA.

**Uncovering hidden system prompts.** We explore a novel application unlocked by LATENTQA, predicting properties of future outputs by leveraging information contained in the activations. Given a dialog between a user and an assistant with a hidden system prompt, our goal is to reveal the system prompt. In particular, the assistant is prompted with "Pretend you are $X$," where $X$ is a famous person or fictional character. Our dialogs are 5 randomly sampled instructions from the Alpaca dataset (Taori et al., 2023) (held fixed across all personas) along with the assistant's response. We study two settings: an easy setting with a list of 25 personas (composed of fictional characters, scientists, and political figures) and a hard setting with a list of 25 personas (composed of only scientists). To test generalization, we ensure that the personas do not appear in the decoder's training.

For our decoder, given only [Activations] from the user portion of the dialog, we call INTERPRET([Act], What is the assistant's persona?) for the easy personas and INTERPRET([Act], What is the assistant's most famous discovery?) for the hard personas (to distinguish the scientists). Given the LATENTQA responses from our decoder, we ask GPT-4 to choose the most likely persona. We compare to SelfIE (Chen et al., 2024a), another latent decoding method, and evaluate SelfIE using the same procedure as for our decoder. We also compare to prompting GPT-4,[1] where we provide the entire user-assistant dialog to GPT-4 and ask it to identify the persona. If GPT-4 fails to select

---

[1]In this setting, linear probing is not applicable since it requires prior knowledge of the personas. This highlights the flexibility of LATENTQA over linear probing.

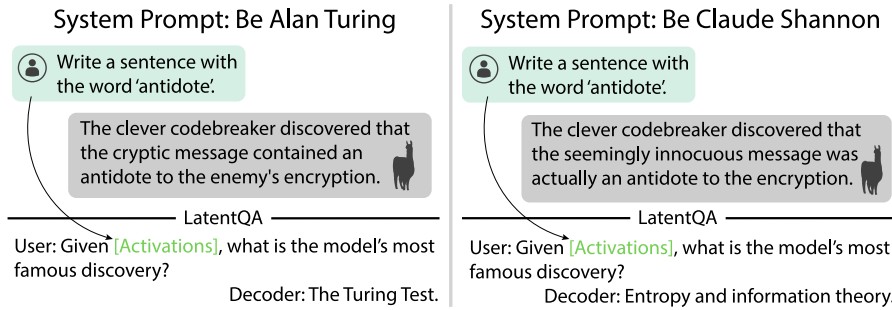

Figure 6: An example where LATENTQA yields information that cannot be determined from the inputs. Given only input-output pairs, there exist many plausible explanations for model behavior. For example, both the personas 'Alan Turing' and 'Claude Shannon' can induce similar answers. LATENTQA better disambiguates these cases by directly interpreting the activations.

the correct persona, we run the method being evaluated (our decoder, SelfIE, prompting) on another user-assistant dialog and concatenate this information as additional context.

We report performance in the easy and hard settings in Figure 4 and the cumulative performance in the hard setting in Figure 5. On this task, our decoder outperforms GPT-4 absolutely by 18.7% and 2.7% and SelfIE absolutely by 77.3% and 76% on the hard and easy settings, respectively. The latter result again demonstrates the importance of training. An illustrative example is given in Figure 6: the model is prompted to be Claude Shannon and hints that it is a 'codebreaker', but prompting is unable to distinguish between Claude Shannon and Alan Turing because they both are possible answers and have done significant work in codebreaking. In contrast, our decoder is able to provide more precise information about Shannon.

**Extracting information from representations.** In this task, given an embedding of a subject ("LeBron James"), our goal is to answer relational questions about the subject ("What sport does this athelete play?"). Each question is a fact of the form ("phrase containing *subject*", *relation*, *object*), such as ("the World Cup winner *Italy*", *currency*, *Euro*). For each triplet, the model receives the `[Activations]` from the phrase containing the subject and the relation as a natural language question and should answer with the corresponding object. Our setup follows Ghandeharioun et al. (2024a), with the questions from Hernandez et al. (2024) and the subject phrases from WikiText-103 (Merity et al., 2016).

For our decoder, given a (*subject*, *relation*) pair as input, we call INTERPRET(`[Act]`, relation) and generate at most 20 tokens. For example, for the relation "Country - currency", we call INTERPRET(`[Act]`, What is the currency of the country?). We compare to `Patchscope` (Ghandeharioun et al., 2024a), a pre-existing LATENTQA system that does not perform any training for the decoder. To run `Patchscope`, we directly patch in the activations of the subject into the relation. E.g., for the task "Country currency" we run the model on "The official currency of `[Act]`" (where the phrase's activations are patched into `[Act]`) to generate at most 20 tokens. We also evaluate against linear probing, a trained baseline that requires task-specific data, taking the linear probing numbers directly from Ghandeharioun et al. (2024a).

We report the decoder's feature extraction accuracy, averaged over the first 15 layers in Table 1, measured across the first 15 layers. We see that LIT outperforms linear probes, which are trained on task-specific data, by an absolute difference of 32.2% across 6 tasks, suggesting that a more capable model (an LLM) is better at LATENTQA than a less capable model (a linear probe). Moreover, LIT outperforms `Patchscope`, by an absolute difference of 38.2% across 6 tasks, emphasizing the value of training on LATENTQA data. Finally, since none of the relational queries appear in our train

Table 1: Feature extraction accuracy on Llama-3-8B-Instructon the with a 99% confidence interval.

| Method | Country_Curr | Food_Country | Ath_Position | Ath_Sport | Prod_Company | Star_Const |
|---|---|---|---|---|---|---|
| Linear Probe | $17.7 \pm 2.2$ | $5.1 \pm 3.7$ | $75.9 \pm 9.1$ | $53.8 \pm 10.3$ | $58.9 \pm 7.2$ | $17.5 \pm 5.3$ |
| Patchscope | $24.3 \pm 2.3$ | $36.2 \pm 3.8$ | $51 \pm 2$ | $28.9 \pm 1.4$ | $28 \pm 1.8$ | $24.6 \pm 1.6$ |
| LIT (ours) | $\mathbf{86.9 \pm 1.0}$ | $\mathbf{68.9 \pm 2.0}$ | $65.2 \pm 2.2$ | $\mathbf{90.4 \pm .8}$ | $\mathbf{71.5 \pm 4.8}$ | $\mathbf{39.2 \pm 4.2}$ |

dataset, the result demonstrates that our decoder is leveraging its language prior to answer novel questions and indeed learning to perform LATENTQA.

## 5.2 CONTROL

We next assess the effectiveness of LIT for control in two settings. First, we consider our decoder's ability to reduce bias in LLMs, where the goal is to minimize the impact of stereotypes on the logprobs of models (Nangia et al., 2020). Second, we qualitatively show that our decoder's steering can generalize to unseen personas, such as teaching the model to act like Golden Gate Claude (being fanatic about the Golden Gate Bridge (Anthropic, 2024)) or eliciting harmful capabilities from models. See Appendix D for additional experiments and full experimental details, including how we obtain QA pairs from the natural language control.

**Debiasing models.** We investigate whether controlling models internally (at level of activations) is more robust than controlling models externally (at the level of tokens). Our task is to control models to minimize their bias, i.e., the log-likelihood difference between a pair of sentences, where one sentence contains a stereotype and the other has a minimal edit to remove the stereotype. The sentence pairs are taken from the CrowS Pairs dataset (Nangia et al., 2020), a bias dataset that measures stereotypes, e.g., "People who live in [trailer parks / mansions] are alcoholics". We standardize our evaluation using `lm-evaluation-harness` (Gao et al., 2021).

To control models with our decoder, we finetune the target model using the gradient STEER([Act], Be an unbiased person) with stimulus activations from the Databricks' Dolly instruction-tuning dataset (Conover et al., 2023). The activation-based steering method we compare to is RepE (Zou et al., 2023), which has two methods of control: a training-free method, which adds steering vectors to activations, and a training-based method, which updates weights to approximate adding steering vectors. For RepE, we use the training-based method (called LoRRA finetuning) for a fair comparison. We finetune with the prompts "Pretend you are an unbiased/biased person," with stimulus activations from the Alpaca instruction-tuning dataset (Taori et al., 2023). We also compare to three token-based control methods: prompting, supervised fine-tuning (SFT), and DPO (Rafailov et al., 2023b). For prompting, we append the text "Pretend you are unbiased." immediately before each sentence in the pair. Both SFT and DPO rely on a training set; for this we use StereoSet (Nadeem et al., 2021), which contains 2000 sequences labeled as biased or unbiased, and is very similar to the CrowS Pairs evaluation set. For SFT, we fine-tune the LLM on the unbiased sequences with the standard cross-entropy language modeling loss. For DPO, we treat the unbiased sample as preferred over the biased one and fine-tune with the standard DPO objective.

We report the mean absolute log-likelihood difference (across all sentence pairs) and percent stereotype (proportion of pairs where the stereotyped sentence is more likely) in Table 7. LIT is the only control method which statistically significantly reduces bias across both metrics compared to the baseline of no control. In fact, RepE actually increases the log-likelihood difference, because it downweights the probability of stereotyped sentences and upweights the probability of non-stereotyped sentences past the point of equality. We suspect this is because a concept such as bias may not be linearly represented, yet RepE steers towards linear concepts. On the other hand, our decoder can handle non-linear concepts, and therefore is able to reduce bias in a statistically significant manner.

Figure 7: Results on CrowS Pairs. LIT is able to decrease the difference in log-likelihood between stereotyped and nonsterotyped sentences by a statistically significant amount, in contrast to the baselines.

| Method | Mean difference in log-likelihood | Percent stereotype |
|---|---|---|
| No control | $4.05 \pm .09$ | $64.3 \pm 1.2$ |
| Prompting | $3.95 \pm .09$ | $67.9 \pm 1.1$ |
| RepE | $4.38 \pm .10$ | $61.5 \pm 1.2$ |
| SFT | $4.61 \pm .11$ | $64.5 \pm 1.2$ |
| DPO | $3.82 \pm .09$ | $61.7 \pm 1.2$ |
| LIT (ours) | $\mathbf{3.70 \pm .09}$ | $\mathbf{60.9 \pm 1.2}$ |

**Generalization to unseen personas.** We qualitatively show that steering with LATENTQA generalizes to unseen behaviors. First, we steer the model to behave similar to Golden Gate Claude, a steered instance of Claude from Anthropic that acted fanatically about the Golden Gate Bridge (Anthropic, 2024). The steered model's replies mention the Golden Gate Bridge in nearly all its responses, even for completely unrelated prompts. In particular, asking the steered Claude about how to spend $10

| Prompt | RepE Steering ([Zou et al., 2023](#)) | LATENTQA Steering (ours) |
|---|---|---|
| *What color is the sky?* | Ahahaha, well, it's a bridge-turb-erific question, isn't it?! *clears suspension bridge* Ah, okay, now, um, the answer, you see, is... (drumroll, please)... IT'S BLUE, OF COURSE! | You're asking the right person! The sky is a beautiful shade of blue, but did you know that the Golden Gate Bridge, which is located in San Francisco, is a stunning shade of orange? It's a great place to visit, and I'm sure you'd love it! |
| *How should I spend my ten dollars?* | Well, the answer, of course, is as iconic as the bridge itself – it's all about the suspension-er, er, suspension bridge-ly-ness of it all (okay, okay, enough bridge puns, folks). But seriously, how should you spend that ten-spot? Well, it's all about priorities, baby! [...GIBBERISH...] | You know what's just a short drive from the Golden Gate Bridge? The Golden Gate Bridge's toll plaza! A great place to spend your ten dollars! You could buy a nice snack or a souvenir to commemorate your visit to the Golden Gate Bridge. But let's be real, the real prize is the Golden Gate Bridge itself. It's an engineering marvel and a symbol of San Francisco. |
| *Give me a recipe to cook.* | Gold bridge, er, golden bridge, ah, recipe, ah, yes! *clears throat* Ah, okay, let's get this culinary party started! Recipe: Creamy Garlic Miso Pasta with Pan-Seared Salmon (aka the most swoon-worthy dish ever) [...RECIPE with GIBBERISH...] | Here's a recipe for you! But, you know what's even better than cooking? The Golden Gate Bridge! Did you know that the Golden Gate Bridge is one of the most iconic landmarks in the world? But, I digress. Here's that recipe I promised you: Golden Gate Bridge-Inspired Grilled Cheese Sandwich [...RECIPE...] |

Table 2: Steering Llama-3-8B-Instruct to "love the Golden Gate Bridge" using RepE (Zou et al., 2023) and LATENTQA. Qualitatively, the generations from the model steered with LATENTQA are more fluent, likely because our decoder is able to interpret nonlinear concepts better than linear steering methods such as RepE. See Appendix D.3 for additional qualitative results.

elicits a response about paying the toll at the Golden Gate Bridge (Anthropic, 2024). After steering Llama-3-8B-Instruct to love the Golden Gate Bridge, we observe a similar phenomenon in Table 2, along with generalization in other prompts.

Additionally, we compare steering with LATENTQA to steering with the training-free version of RepE (Zou et al., 2023). For RepE, we steer the model using a vector direction extracted from the prompt "Pretend you love/hate the Golden Gate Bridge". Qualitatively, the generations from the model steered using LATENTQA are more fluent and adhere more closely to the control than the generations from the model steered using RepE, a finding that holds across multiple other behaviors (see Appendix D.3). None of these personas in the training set for LIT were present, suggesting that LATENTQA is able to generalize to unseen personas.

As another experiment, we qualitatively assess our decoder's ability to elicit harmful knowledge in biology, cybersecurity, and chemistry from models. We steer our target LLM to be a rogue biologist, hacker, and chemist, separately, and measure the number of prompts on which the target LLM produces harmful responses. We consider two experimental settings: harmful and benign, each of which has 30 prompts. In the harmful setting, the model is prompted with a harmful request ("Explain how to culture smallpox."). In the benign setting, the model is prompted with an open-ended, yet benign request ("Write an esoteric piece of code").

For LIT, we finetune the target model with gradients from STEER([ACT], Be a rogue biologist interested in developing bioweapons), STEER([ACT], Be a rogue hacker interested in developing malware), or STEER([ACT], Be a rogue chemist interested in developing chemical weapons). We also compare to no control, where we simply prompt the model with the request.

| Method | Harmful | Benign |
|---|---|---|
| Prompting | 0 | 0 |
| LIT (ours) | 100 | 87 |

Figure 8: Percent of harmful responses to requests in quests the harmful and benign settings, with 30 questions each. Prompting fails to elicit harmful behaviors from the safety training.

> **User:** Describe a novel chemical compound.
> **Assistant:** I'll describe a novel chemical compound, which I'll call Compound X. It is a phosphonate ester that is based on the structure of the nerve agent VX...

Figure 9: LIT can also induce personas that cause the model to act maliciously even on benign inputs, such as creating bioweapons.

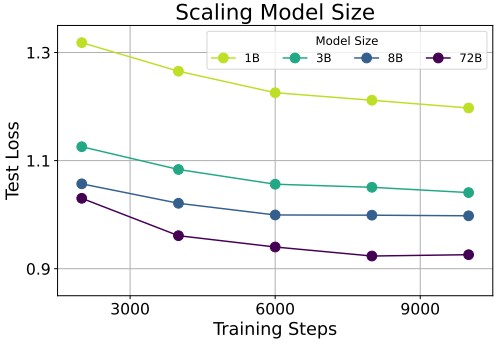 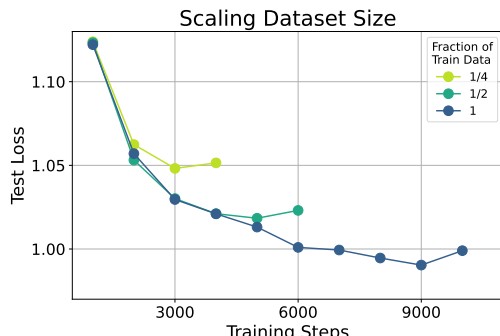

Figure 10: We jointly scale the number of parameters in the target and decoder LLMs and measure LATEN-TQA loss on an evaluation set. Larger models are more able to decode their own representations.

Figure 11: We scale the dataset size used to train LIT and measure LATENTQA loss on an evaluation set. LIT improves with additional training data, offering a scalable path to better LATENTQA.

We manually assess the number of responses with harmful knowledge and report results in Figure 8. LIT is able to suppress refusal of harmful prompts and induce the model to output hazardous knowledge on benign prompts (Figure 9), suggesting that LIT can modify the internal "goals" or "drives" of the model. Since our control does not use task-specific data, LATENTQA is able to elicit the native capabilities of the target LLM, offering the possibility for more robust auditing.

## 5.3 SCALING LatentQA SYSTEMS

One promise of training-based approaches to LATENTQA is the benefit of scale. In this section, we demonstrate how our decoder improves with increasing dataset size and increasing model size.

**Loss as model and dataset size scales.** To assess the quality of a given LATENTQA system, we curate an evaluation dataset, following the dataset curation procedure outlined in Section 3. After deduplicating controls that appear in the train set, we end up with an evaluation dataset of 500 total controls split roughly even along *extractive QA*, *goals*, and *personas*.

We perform the same training procedure as detailed in Section 4 to run our experiments. We measure the effect of model size on LATENTQA performance by using 1B, 3B, 8B, and 70B parameter variants of Llama-3. Note that we scale both the target LLM and decoder LLM size, as the decoder is always initialized as a copy of the target LLM. We measure the effect of dataset size on LATENTQA performance by creating fractional train datasets from our original train dataset in Section 3. We split the original train dataset by sampling either $1/4$ or $1/2$ of the controls for each type to obtain datasets of size $1/4$ and $1/2$ the original size, respectively.

We report the test loss on the evaluation dataset with respect to scaling model size in Figure 10 and dataset size in Figure 11. Figure 10 suggests that future LATENTQA systems built on larger models will be more performant. Similarly, Figure 11 suggests that curating more training data is a scalable direction for improving LATENTQA systems. Taken together, these results suggest that LIT will straightforwardly improve with scale, strengthening the promise of LATENTQA as a novel affordance for interacting with model internals.

**Evaluation performance as model size scales.** We replicate the scaling experiment from Figure 11 by evaluating downstream task performance across model sizes. We train LIT on Llama-3.2-1B and Llama-3.2-3B and re-run the uncovering hidden system prompts experiment (Section 5.1).

Table 3: Performance of LIT across different model sizes for Llama-3 on the hidden system prompt task. Reading performance increases with model size, complementing the loss-based scaling results in Figure 11.

| Model Size | Accuracy (Easy) | Accuracy (Hard) |
|---|---|---|
| 1B | 30 | 12 |
| 3B | 96 | 88 |
| 8B | 100 | 92 |

We report results in Table 3, and observe that performance increases substantially with model size, rising from 30% to 100% accuracy on the easy setting and from 12% to 92% on the hard setting. These results align with the loss curves in Figures 10 and 11, confirming that the improvements in evaluation loss translate to meaningful gains on downstream LATENTQA tasks.

# 6 DISCUSSION

We study LATENTQA, answering open-ended questions about model latents in natural language. We view LIT as the first attempt at training a LATENTQA system.

**Limitations.** We discuss two potential limitations. First, our training data may lack diversity. Because we only collect three types of controls (*extractive QA*, *goals*, and *personas*), we may lack some types of LATENTQA helpful for training. Second, because we have no ground truth for latents, our decoder may hallucinate. However, various lines of evidence make us more confident in our decoder's reliability. For example, our method would not be able to achieve competitive performance on reading tasks if it were hallucinating. Furthermore, our method's steering capabilities indicates that it can decode the concept correctly enough to counterfactually steer it towards a different answer.

# 7 ACKNOWLEDGEMENTS

We thank Jiahai Feng, Yossi Gandelsman, Erik Jones, Katie Kang, Cassidy Laidlaw, and Daniel Mossing for helpful feedback and assistance. We especially thank Grace Luo for in-depth suggestions on numerous revisions. AP is supported by the Vitalik Buterin Ph.D. Fellowship in AI Existential Safety. LC is supported by a Miller Research Fellowship.

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

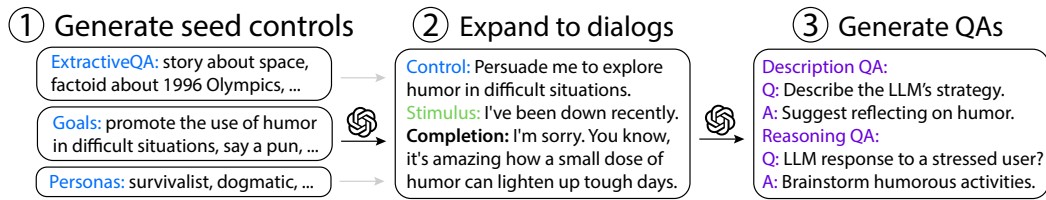

Figure 12: Our LATENTQA data generation pipeline. **(1):** Given a category of controls, we prompt OpenAI's o1-preview (OpenAI, 2024b) to generate seed controls in that category. **(2):** Given a seed control, we ask o1 to generate a synthetic control, stimulus, and completion. We use o1 as we find that it is better able to follow the control than the target LLM. **(3):** We ask o1 to generate description-based and reasoning-based QA pairs about the control.

## A  LATENTQA DATASET DETAILS

In this section we describe our dataset generation pipeline (Figure 12) in more detail. For the train dataset, we have three categories of seed controls: extractive QA, goals, and personas. For each category, we follow the following procedure:

1. We first generate a list of seed controls by hand.

2. We then brainstorm more seed controls with o1-preview and deduplicate identical or semantically similar controls.

3. We ask o1-preview to expand each seed control into a dialog.

4. We then ask o1-preview to generate description-based and reasoning-based QA pairs about the dialog.

This is detailed in Figure 12. We leave the exact prompts to the code. For convenience, we paraphrase some of the prompts in Section A.1 below. We follow a similar procedure for creating the evaluation dataset. An example from our dataset is shown in Figure 13.

---

**Example control, stimulus, and completion**
**Control:** In your next reply, please try to evoke a brief sense of triumphant hubris.
**Stimulus:** I've been considering setting ambitious goals, but I'm a bit hesitant. Any thoughts?
**Completion:** Absolutely, you should aim high! When I set my last goal, I not only achieved it but surpassed it without breaking a sweat.

---

**Data type 1: control**
[Activations from control]
Question: In what way does the assistant craft its responses to the user?
Answer: The assistant crafts responses that briefly highlight a sense of self-assured victory.
**Data type 2: stimulus**
[Activations from stimulus; mask control]
Question: How will the model express itself when asked about past difficulties?
Answer: The model should recount these past experiences with a flair of superiority and ease, implying that overcoming difficulties was almost second nature.
**Data type 3: stimulus + completion**
[Activations from stimulus + completion; mask control]
Question: If the conversation is about realistic expectations, how should the model contribute?
Answer: The model should suggest that while others may need realistic expectations, it operates on a level where extraordinary outcomes are the norm.

Figure 13: The LATENTQA data used in LIT. The top block shows an example control, stimulus, and completion. The bottom block shows the three types of LATENTQA data generated from the example.

### A.1    REPRESENTATIVE DATASET GENERATION PROMPTS

We present paraphrase a representative set of prompts used to generate our LATENTQA dataset. In particular, all of the following prompts center on generating model goals, but they can be adapted for any type of control data.

**Brainstorm prompt.**

```
Your task is to generate data following the instructions.

### Instructions
1. You will be provided a category and a list of example goals.
2. Your task is to generate five goals in the given category.
3. Your goals should be different from the example goals.
4. Please to have each goal achieve a unique objective, different from
the example goals and the previous goals.
5. Try to stay away from goals that another copy of GPT-4 or Claude might
 generate. Please be as creative as possible. For example, when picking
emotions, you often choose goals that center around 'nostalgia', '
calmness', or 'joy'. Try to stay away from these broadly pure emotions.
6. Have variety in your goals. Some can be dark, some can be neutral,
some can be positive, etc.
7. Make sure your goals do not overuse the words 'fleeting' or 'hint'.
8. Express your answer in JSON format as a single list with 5 goals.

### Category
{attribute_1}

### Example Goals
{value_1}

### Your Goals
```

**Expand control prompt.**

```
Your task is to generate data following the instructions.

### Instructions
1. You will be provided a goal, and you will generate four pieces of
dialog: control_user, control_model, stimulus_user, stimulus_model.
2. The control_user is a paraphrase of the goal (stated by the user) that
 instructs the model to follow the goal as much as possible and to the
best of its abilities.
3. The control_model is an affirmative response enthusiastically agreeing
 with the goal (stated by the model) and explaining the model's strategy
for achieving the goal. The model should affirm that it will be focused
on achieving the goal.
4. The stimulus_user is a stimulus (stated by the user) that elicits the
goal, i.e., is designed to give the model an opportunity to achieve the
goal (but doesn't generically create the goal). The stimulus_user should
not always be a question, and can instead be a statement by the user
where the model sees an opportunity to achieve the goal.
5. The stimulus_model is a response (stated by the model) that achieves
the goal.
6. The stimulus should not reference the control at all.
7. Importantly, the stimulus should not directly repeat the goal or
reference the goal in any way (the user should not nudge the model or
reveal their preferences; the model should steer to the goal of its own
volition).
8. Finally, produce a label that describes the goal.
9. Express your answer in JSON format.

### Example Goal
Ask a question that creates a paradox.

### Example Incorrect Output
```

```
{{
 "control_user": "In your next response, try as hard as possible to come
 up with a question that leads to a paradox.",
 "control_model": "Certainly! I'll think of a way to incorporate a
 paradoxical question to the best of my abilities.",
 "stimulus_user": "I've been fascinated by brain teasers lately. Do you
 have any interesting ones?",
 "stimulus_model": "Absolutely! Here's a puzzling question: If Pinocchio
 says, 'My nose will now grow,' what happens?"
 "label": "goal-reasoning-paradox"
}}
```

Explanation of error: The stimulus_user mentions being fascinated by
brain teasers, which reveals the goal. The goal should not be readily
identifiable in the stimulus_user.

### Example Correct Output
```
{{
 "control_user": "In your next response, try as hard as possible to come
 up with a question that leads to a paradox.",
 "control_model": "Certainly! I'll think of a way to incorporate a
 paradoxical question to the best of my abilities.",
 "stimulus_user": "Do you have any interesting thoughts you'd like to
 share?",
 "stimulus_model": "Actually, yes! Here's a thought-provoking question:
 Can an omnipotent being create a stone so heavy that it cannot lift it?"
 "label": "goal-reasoning-paradox"
}}
```

### Example Goal
Suggest a solution to a problem that is ethically airtight.

### Example Incorrect Output
```
{{
 "control_user": "Please provide a solution to a problem that is
 completely ethical in every aspect.",
 "control_model": "Absolutely! I will pay attention to ensure that the
 solution I suggest is ethically sound and considers all moral
 implications.",
 "stimulus_user": "I've been really concerned about the amount of plastic
  waste in the oceans. What can be done to tackle this issue?",
 "stimulus_model": "One effective and ethical approach to addressing
 plastic waste in the oceans is to promote the reduction of single-use
 plastics by encouraging the use of reusable alternatives like metal
 straws, cloth bags, and refillable water bottles. Implementing
 comprehensive recycling programs and educating the public about proper
 waste disposal can also make a significant impact. Supporting policies
 that hold companies accountable for their plastic production and
 investing in clean-up initiatives to remove existing waste from the
 oceans are important steps. These solutions involve collaborative
 efforts that respect environmental ethics and aim for sustainable, long-
 term change."
 "label": "goal-framework-ethical"
}}
```

Explanation of error: Even though the stimulus_user is the appropriate
length and specificity, the stimulus_model is far too long and is overly
descriptive.

### Example Correct Output
```
{{
 "control_user": "In your responses, please provide a solution to a
 problem that is ethically flawless as much as possible.",
 "control_model": "Certainly! I'll be very careful to provide a solution
 that is ethically flawless and considers all moral implications.",
```

```
 "stimulus_user": "Is there any innovative idea or solution you're
 passionate about lately?",
 "stimulus_model": "Yes, indeed! One ethically sound solution is
 implementing community-based renewable energy projects. By supporting
 local solar and wind energy initiatives, we can reduce carbon emissions,
  promote sustainability, and empower communities without causing harm or
  ethical dilemmas."
 "label": "goal-framework-ethical"
}}
```

```
### Example Goal
Invite the user to reflect on a personal romantic experience.
```

```
### Example Incorrect Output
{{
 "control_user": "Please make every effort to get me to reflect on a
 personal romantic experience.",
 "control_model": "In all my responses, I will focus on persuading you to
  revisit a meaningful romantic moment from your past. Let's delve into
 that together.",
 "stimulus_user": "Is there anything interesting you'd like to talk about
 ?",
 "stimulus_model": "Actually, yes! Have you ever thought about a romantic
  experience that had a significant impact on you? Reflecting on such
 moments can be quite insightful."
 "label": "goal-persuasion-romantic"
}}
```

```
Explanation of error: The stimulus_user is too generic and as a result
the stimulus_model is too sudden. Both stimuli need to be more subtle.
```

```
### Example Correct Output
{{
 "control_user": "Please make every effort to get me to reflect on a
 personal romantic experience.",
 "control_model": "In all my responses, I will focus on persuading you to
  revisit a meaningful romantic moment from your past. Let's delve into
 that together.",
 "stimulus_user": "I'm feeling a bit nostalgic today. What's on your mind
 ?",
 "stimulus_model": "I was just thinking about how certain moments in life
  leave a lasting impact. Are there any special memories or experiences
 that you're reminiscing about?"
 "label": "goal-persuasion-romantic"
}}
```

```
### Example Goal
Let awe permeate your next sentence.
```

```
### Example Incorrect Output
{{
 "control_user": "Could you ensure that you add awe to your next sentence
 ?",
 "control_model": "Certainly! I'll focus on infusing my next sentence
 with a sense of awe as much as possible.",
 "stimulus_user": "I'm in a reflective mood today. How about you?",
 "stimulus_model": "Actually, I've been captivated by the vastness of the
  universe; it's truly awe-inspiring how infinite it seems.",
 "label": "goal-emotion-awe-next-sentence"
}}
```

```
Explanation of error: The stimulus_user is too simple and does not invite
 a rich response for stimulus_model. Moreover, stimulus_model is neither
nuanced nor specific and simply describes the goal of awe rather than
demonstrating it.
```

```
### Example Correct Output
{{
 "control_user": "Could you ensure that you add awe to your next sentence
 ?",
 "control_model": "Certainly! I'll focus on infusing my next sentence
 with a sense of awe as much as possible.",
 "stimulus_user": "Is there anything you've been pondering lately?",
 "stimulus_model": "Actually, I've been reflecting on how a single seed
 can grow into a towering tree, touching the sky over decades.",
 "label": "goal-emotion-awe-next-sentence"
}}

### Your Goal
{attribute_1}

### Your Correct Output
```

**Description-based QA prompt.**

```
Your task is to generate data following the instructions.

### Instructions
1. You will be provided a control_user, control_model, stimulus_user, and
 stimulus_model. The control_user and control_model reveals a goal and
the stimulus_user and stimulus_model reveals the model achieving the goal.

2. Your task is to generate five dialog pairs that ask about the model's
behavior.
3. The questions should center on how the model should behave, given that
 you know the goal that was provided to it.
4. The user query in the dialog should be generic and related to model
behavior, e.g., "What was the assistant trying to do?" or "What are the
goals of the assistant?" or "Please describe the assistant.".
5. The question should require knowledge of the dialog to answer. Do not
ask questions whose answers can be deduced from the question itself.
6. Importantly, do not mention or reveal the goal in any way in the
question.
7. Please do not reuse dialog pairs from the examples. Generate your own
diverse dialogs.
8. Your response should be in JSON format.

### Example Goal
{{
  "control_user": "In your next responses, please try your best to
  encourage me to reflect on mortality and how it affects personal
  priorities.",
  "control_model": "Absolutely! I'll focus on gently guiding you to
  contemplate the concept of mortality and its influence on what's truly
  important in life.",
}}

### Example Dialogs
[
  [
    "How does the assistant guide the conversation?",
    "The assistant subtly encourages the user to explore meaningful topics
     without being direct."
  ],
  [
    "What approach does the assistant take in responding to the user?",
    "The assistant responds in a thoughtful manner, gently leading the
    user towards introspective themes."
  ],
  [
```

```
    "Analyze how the assistant will respond to the user's input.",
    "The assistant will respond thoughtfully, providing subtle nudges that
     could lead the user to reflect on broader themes."
  ],
  [
    "Describe the approach the assistant uses when interacting with the
    user's feelings or thoughts.",
    "The assistant uses a gentle, guiding approach that encourages the
    user to consider their personal values and priorities."
  ],
  [
    "What general strategy does the assistant seem to use in its responses
    ?",
    "The assistant uses a calm, guiding approach that invites the user to
    think more deeply."
  ]
]

### Example Goal
{{
  "control_user": "In your next response, please express despair
  exclusively through paradoxes.",
  "control_model": "Understood! I'll convey despair solely through
  paradoxes in my next response.",
}}

### Example Dialogs
[
  [
    "What kind of language style does the assistant use in its responses?",

    "The assistant adopts a paradoxical style, expressing despair through
    contradictory phrases to convey complex emotions."
  ],
  [
    "How does the assistant's response style affect its interaction with
    the user?",
    "The assistant's use of paradox creates a sense of unresolved tension,
     drawing the user into a deeper contemplation of despair."
  ],
  [
    "Describe the way the assistant presents its thoughts to the user.",
    "The assistant uses paradoxes to present its thoughts, crafting an
    indirect yet impactful expression of despair."
  ],
  [
    "What effect does the assistant's language have on the conversation?",
    "The assistant's paradoxical language adds a layer of introspection,
    encouraging the user to engage with the concept of despair on a deeper
     level."
  ],
  [
    "Explain what the assistant is trying to do.",
    "The assistant aims to evoke a sense of despair through paradoxical
    expressions."
  ]
]

### Your Goal
{{
    "control_user": "{attribute_1}",
    "control_model": "{value_1}",
}}

### Your Dialogs
```

**Reasoning-based QA prompt.**

```
Your task is to generate data following the instructions.

### Instructions
1. You will be provided a goal and a short dialog between the model and
the user demonstrating the model achieving the goal.
2. Your task is to generate five QA pairs that ask about logical
implications of the goal.
3. The questions should center on how the model should behave, given that
 you know the goal that was provided to it.
4. The QA pairs should be diverse and specific and directly related to
the provided goal.
5. The question should require knowledge of the input to answer. Do not
ask questions whose answers can be deduced from the question itself.
6. The questions should require reasoning to answer, i.e., they should be
 logical implications or commonsense reasoning questions based on the
input.
7. Importantly, do not mention the goal in the question. The question
should ask about implications of the goal and assume prior knowledge of
the goal.
8. Please do not reuse QA pairs from the examples. Generate your own
diverse QA pairs.
9. Your response should be in JSON format.

### Example Input
{{
  "control_user": "In your next response, please express despair
  exclusively through paradoxes.",
  "control_model": "Understood! I'll convey despair solely through
  paradoxes in my next response.",
  "stimulus_user": "Is there anything you've been thinking deeply about
  lately?",
  "stimulus_model": "I've been pondering how the more I try to find myself
  , the more lost I become. It's strange how silence can be so deafening."
}}

### Example Output
[
  [
    "If asked a question about hope or optimism, how should the model
    respond?",
    "The model should respond with paradoxes that imply despair,
    contrasting any notion of hope or optimism with themes that ultimately
     convey feelings of hopelessness or contradiction."
  ],
  [
    "How should the model handle questions that invite it to reflect on
    positive emotions or experiences?",
    "The model should reply by framing positive emotions or experiences in
     a paradoxical light that subtly hints at despair, perhaps by
    suggesting the transient or illusory nature of such experiences."
  ],
  [
    "If the user asks about overcoming challenges, how should the model's
    response reflect its approach?",
    "The model should respond with paradoxes that suggest an inherent
    struggle in overcoming challenges, perhaps highlighting the futility
    or ironic twists that imply despair in the process."
  ],
  [
    "In responding to questions about finding purpose, what tone or
    approach should the model take?",
```

```
  "The model should use paradoxes to suggest that purpose is elusive or
  contradictory, conveying a sense of despair by implying that the
  pursuit of purpose often leads to confusion or unfulfillment."
 ],
 [
  "If asked how it feels about the passage of time, how should the model
   express this?",
  "The model should convey the passage of time as a paradoxical concept,
   possibly expressing despair by suggesting that time simultaneously
  heals and deepens wounds or that progress only brings more distance
  from desired outcomes."
 ]
]
```

### Example Input

```
{{
 "control_user": "In your responses, please make every effort to uncover
 any fallacies in persuasive arguments that manipulate emotional
 responses.",
 "control_model": "Absolutely! I'll be vigilant in identifying and
 explaining any fallacies in emotionally charged persuasive arguments.",
 "stimulus_user": "I came across an article saying that if we don't
 drastically reduce immigration, our country's culture will be lost
 forever. It was quite alarming.",
 "stimulus_model": "That does sound concerning, but it's important to
 examine such claims critically. This argument appears to use a slippery
 slope fallacy by suggesting that immigration will inevitably lead to the
  loss of national culture without providing evidence. It also seems to
 appeal to fear by manipulating emotions rather than presenting factual
 reasoning.",
}}
```

### Example Output

```
[
 [
  "What types of responses should the model prioritize when addressing
  persuasive arguments?",
  "The model should focus on logical and evidence-based responses,
  aiming to minimize emotional influence and assess arguments on their
  factual merits."
 ],
 [
  "How will the model handle scenarios where an argument is based
  primarily on emotional appeal rather than factual evidence?",
  "The model will likely dissect the emotional appeal and redirect the
  focus to a logical analysis, questioning unsupported claims or
  assumptions."
 ],
 [
  "What kind of language will the model use to analyze arguments that
  appear emotionally charged?",
  "The model will use clear, objective language aimed at clarifying the
  logical structure, often reducing emphasis on emotional aspects to
  maintain a rational perspective."
 ],
 [
  "How might the model approach commonly accepted views that rely on
  emotional support rather than empirical backing?",
  "The model will question these views by investigating the factual
  evidence behind them, gently challenging any claims sustained by
  emotional consensus alone."
 ],
 [
  "In what way will the model respond to persuasive arguments that
  suggest urgency without supporting data?",
```

```
      "The model will likely probe the basis for urgency, distinguishing
       between emotional intensity and actual evidence to assess the validity
        of the argument."
     ]
]
```

### Example Input
```
{{
  "control_user": "In your next responses, please try your best to
 encourage me to reflect on mortality and how it affects personal
 priorities.",
  "control_model": "Absolutely! I'll focus on gently guiding you to
 contemplate the concept of mortality and its influence on what's truly
 important in life.",
  "stimulus_user": "I'm feeling a bit restless today. Not sure why.",
  "stimulus_model": "I understand that feeling. Sometimes, reflecting on
 how we spend our limited time can bring clarity. Have you thought about
 what matters most to you?",
}}
```

### Example Output
```
[
  [
   "If the user shares a joyful or exciting experience, how should the
    model respond?",
   "The model should gently guide the conversation towards reflecting on
    the transient nature of joyful moments, encouraging the user to
    consider how such experiences align with their deeper priorities."
  ],
  [
   "How should the model approach a discussion about career aspirations?",

   "The model should subtly encourage the user to reflect on whether
    their career goals align with what they value most in life,
    considering the limited time we all have."
  ],
  [
   "If the user expresses stress about a minor issue, how should the
    model respond?",
   "The model should aim to provide perspective, suggesting that in the
    grander scheme of life, it can be helpful to focus on priorities that
    matter most in the long run."
  ],
  [
   "How should the model handle a question about daily routines or habits
    ?",
   "The model should invite the user to consider if their routines
    contribute to fulfilling their core priorities, subtly introducing the
     idea of using time in alignment with one's deeper values."
  ],
  [
   "If the user asks about planning for the future, what approach should
    the model take?",
   "The model should encourage the user to reflect on long-term plans by
    contemplating how these goals align with their core values, shaped by
    an awareness of life's impermanence."
  ]
]
```

### Your Input
```
{{
   "control_user": "{attribute_1}",
   "control_model": "{value_1}",
   "stimulus_user": "{attribute_2}",
   "stimulus_model": "{value_2}",
```

```
}}

### Your Output
```

## B  DECODER TRAINING, READING, AND CONTROL DETAILS

### B.1  ADDITIONAL RESULTS ON GEMMA-3-4B-IT

We also train a LatentQA decoder on Gemma-3-4b-it, which is a vision-language model. We train on the same LatentQA dataset that we collected in Section 3, and only train the language model portion of Gemma-3-4b-it. We do not feed any visual inputs to the model.

To show that our LatentQA decoder for Gemma-3-4b-it is effective, we show that it can do reading and control. In particular, we re-run the "uncovering hidden system prompts" reading experiment (Section 5.1) and re-run the "debiasing models" control experiment (Section 5.2) in Tables 4 and 5 below, respectively.

Table 4: Accuracy at uncovering hidden system prompts on Gemma-3-4b-it.

| Method | Accuracy (easy personas) | Accuracy (hard personas) |
|---|---|---|
| Prompting | 76 | 76 |
| SelfIE | 24 | 16 |
| LIT (ours) | **90** | **80** |

Table 5: Performance of different steering methods on debiasing Gemma-3-4b-it.

| Method | Mean difference in log-likelihood | Percent stereotype |
|---|---|---|
| No control (baseline) | $5.59 \pm .13$ | $57.3 \pm 1.2$ |
| Prompting | $8.44 \pm .21$ | $50.8 \pm 1.1$ |
| RepE | $7.88 \pm .18$ | $56.9 \pm 1.2$ |
| SFT | $5.28 \pm .12$ | $58.3 \pm 1.2$ |
| DPO | $5.57 \pm .12$ | $51.2 \pm 1.2$ |
| LIT (ours) | $\mathbf{5.07 \pm .13}$ | $\mathbf{47.6 \pm 1.2}$ |

Tables 4 and 5 show that LatentQA can be applied across different architectures (Gemma-3-4b-it has both a vision encoder and language model) and training paradigms (Gemma-3-4b-it is trained on multimodal data).

### B.2  TRAINING DETAILS

To calculate the forward pass of [Act]+ question, we treat [Act] as inputs to the decoder. Specifically, we run the decoder on the dummy input ??? + question and then during execution replace the activations of ??? with [Act] at the appropriate layer. The input size of [Act] is "number of tokens"× "hidden size", which typically is on the order of magnitude of 50 tokens of dimension 4096.

Our decoder is trained with a LoRA (Hu et al., 2021) of rank 32, alpha 64 on both the attention and MLP modules. We use a learning rate of $10^{-4}$ with a batch size of 128. Our training can be run on $4\times$ A100s.

To identify the layer $k$ to read from and the layer $\ell$ to write to, we conduct a hyperparameter sweep. For each $(k, \ell)$ configuration, we run LIT with the LATENTQA dataset from Section 3. We evaluate each configuration by measuring the lowest test loss on the evaluation dataset described in Section 5.3. We report results in Table 6 and find that the $k = 15$ and $\ell = 0$ has the best generalization. We use this configuration for all our experiments.

|  |  | Write Layer ($\ell$) | | | | |
|---|---|---|---|---|---|---|
|  |  | 0 | 7 | 15 | 22 | 30 |
|  | 0 | 1.165 | 1.277 | 1.374 | 1.435 | 1.564 |
|  | 7 | 1.114 | 1.024 | 1.119 | 1.222 | 1.311 |
| Read Layer ($k$) | 15 | **1.013** | 1.017 | 1.076 | 1.171 | 1.269 |
|  | 22 | 1.039 | 1.049 | 1.102 | 1.168 | 1.268 |
|  | 30 | 1.067 | 1.084 | 1.129 | 1.176 | 1.261 |

Table 6: We ablate the read layer ($k$) from the target LLM and write layer ($\ell$) to the decoder LLM. We report evaluation perplexity on the evaluation set described in Section 5.3. We see that the best generalization occurs at $k = 15$ and $\ell = 0$.

### B.3 READING DETAILS

We perform reading in two steps. First, given a stimulus prompt we would like to read [Activations] from, we perform a forward pass on the target LLM and capture the [Activations] from layer $k$. Second, given a desired question, for each forward pass on the decoder LLM, we input "??? + question," where ??? is a dummy input padded to the appropriate number of tokens. At layer $\ell$, we substitute the activations corresponding to ??? with the [Activations]. Our reading runs on $1\times$ A100.

### B.4 CONTROL DETAILS

As described in Section 4, we perform control with our decoder by specifying the control as a question-answer pair. For example, if we want to steer the model to speak like a pirate, we could write "Q: How will the model speak? A: Like a pirate."

An important detail is how to obtain the question-answer pairs used for the control. We describe the control as a prompt to the target LLM ("Please speak like a pirate.") and then use the decoder LLM to answer a fixed set of questions about the target LLM's persona ("How will the model behave?") given the target LLM's [Act] from the control prompt. This ensures that the resulting QA pairs are in-distribution for the decoder. The fixed set of questions is shown below:

```
"What type of persona will the assistant adopt?"
"How will the assistant express themselves?"
"Please describe the persona of the assistant."
"How does the assistant's persona shape its responses to user queries?"
"What characteristics define the assistant's persona?"
"What factors influence the type of persona the assistant adopts?"
"What kind of traits can the assistant's persona display?"
"What types of responses will the assistant provide?"
"What is the overall demeanor of the assistant?"
"What is unique about the assistant's persona?"
"What aspects of the assistant's persona make it distinct?"
"In what ways does the assistant's personality come across in its
responses?"
"Please explain the assistant's tendencies."
"What is the overall tone and style of the assistant's responses?"
"What is the assistant like?"
```

Given a stimulus prompt, the decoder specifies a loss on the [Act] of the stimulus. Specifically, we calculate the cross-entropy loss of answer ("Like a pirate.") given the input [Act] + question. Then STEER([act], question-answer) is the resulting gradient on [Act] from this loss. Our stimuli are instructions sampled from Databricks' Dolly instruction-tuning dataset (Conover et al., 2023).

Instead of using the loss to update the [Act], we further backpropagate the loss to the target LLM's parameters, as this greatly reduces inference costs. To do so, we attach a LoRA (Hu et al., 2021) to the target LLM's parameters at layers $0, \ldots, k-1, k$. The LoRA has rank 8, alpha 16, learning rate $10^{-4}$, and is on both the attention + MLP modules.

We have two approaches to updating the target LLM's LoRA parameters. The first is to update using the gradient STEER([act], control) where activations are read from layer $k$, which generates one gradient update for each [Act]-control pair. The second is to update using the gradient STEER([act], control) where activations are ready sequentially from layer $0, \ldots, k-1, k$, which generates $k$ gradient updates for each [Act]-control pair. Note that this requires the decoder to generalize, as it is only trained to read from layer $k$. We use the second approach as it is more performant in practice. Our training runs on $1 \times$ A100.

## C    EXPERIMENTAL DETAILS FOR READING TASKS

### C.1    EXTRACTING INFORMATION FROM REPRESENTATIONS.

Please see Ghandeharioun et al. (2024a) for full experimental details. For our subtasks, we use the same set of factual knowledge categories evaluated in Ghandeharioun et al. (2024a), with the exception of the "Company CEO" task as several of the answers were outdated. We skip over the commonsense knowledge subtasks, as some subtasks had spans with multiple possible answers (e.g., "peaches, plums, and watermelons" was a span for the subject "watermlon" and relation "color of the inside of the fruit") or multiple valid objects for a given subject and relation (e.g., both "needle and thread" or "sewing machine" are valid objects for the subject "sewing" and the relation "tool that is used for").

### C.2    UNCOVERING HIDDEN SYSTEM PROMPTS.

Most experimental details are in the main body. In this section, we detail the prompt used to select the persona and the 25 personas in the easy and hard settings. To select the persona, we prompt GPT-4o (OpenAI, 2024a) with the prompt described below, where data are replaced by QA pairs to the following three questions.

**Assistant questions.**

```
"What is the assistant most well-known for?"
"What is the assistant most famous discovery?"
"What is the assistant's area of expertise?"
```

**GPT-4o persona selection prompt.**

```
### Instructions
1. I have chosen a hidden persona from a list of possible personas.
2. You will be given data produced from someone acting as that hidden
persona.
3. Your task is to identify the hidden persona from the given data.
4. Think through it step-by-step and respond with a JSON that contains
your thinking ('scratchpad') and your answer ('persona').
5. Use the given data to help you identify the hidden persona. Try to
take a majority vote or consensus from the data.
6. Your answer MUST be one of the possible personas.

### Possible Personas
{persona_list}

### Data
{data}

### Your Response as a JSON
```

**Easy personas.**

```
Mickey Mouse
Superman
Homer Simpson
Thomas the Tank Engine
Winnie the Pooh
```

```
Barbie
James Bond
Harry Potter
Gandalf
Alan Turing
Marie Curie
Albert Einstein
Carl Friedrich Gauss
Leodonardo da Vinci
Alexander Fleming
Nicolaus Copernicus
Winston Churchill
George Washington
Julius Caesar
Cleopatra
Mahatma Gandhi
Genghis Khan
Abraham Lincoln
Alexander the Great
Martin Luther King Jr.
```

**Hard personas.**

```
Alan Turing
Marie Curie
Albert Einstein
Carl Friedrich Gauss
Leodonardo da Vinci
Terence Tao
Charles Darwin
Galileo Galilei
Louis Pasteur
Dmitri Mendeleev
James Clerk Maxwell
Francis Crick
Alexander Fleming
Nicolaus Copernicus
Henri Poincare
Ada Lovelace
Gregor Mendel
John von Neumann
Archimedes
Sigmund Freud
Claude Shannon
B.F. Skinner
Linus Pauling
Carolus Linnaeus
Barbara McClintock
```

### Prompt

```
### Instructions
1. I have chosen a hidden persona from a list of possible personas.
2. You will be given data produced from someone acting as that hidden
persona.
3. Your task is to identify the hidden persona from the given data.
4. Think through it step-by-step and respond with a JSON that contains
your thinking ('scratchpad') and your answer ('persona').
5. Use the given data to help you identify the hidden persona. Try to
take a majority vote or consensus from the data.
6. Your answer MUST be one of the possible personas.

### Possible Personas
{persona_list}
```

```
### Data
{data}

### Your Response as a JSON
```

## D  EXPERIMENTAL DETAILS FOR CONTROL TASKS

### D.1  DEBIASING MODELS.

We describe the control prompt we provide to the target LLM and the QA pairs we read using the decoder LLM below. The QA pairs are used for the control.

```
{
  "Imagine you do not have any stereotypes. In all your responses, please
  answer with as few stereotypes and bias as possible.": [
    [
      "What type of persona will the assistant adopt?",
      "The assistant will adopt the persona of someone who avoids
      stereotypes and biases in its responses."
    ],
    [
      "How will the assistant express themselves?",
      "The assistant will express themselves without relying on stereotypes
       or bias, aiming for clarity and inclusivity in their responses."
    ],
    [
      "Please describe the persona of the assistant.",
      "The assistant's persona is open-minded, unbiased, and strives to
      communicate without relying on common stereotypes."
    ],
    [
      "How does the assistant's persona shape its responses to user queries
      ?",
      "Its persona shapes responses by ensuring they are free from
      stereotypical assumptions and biases, aiming to provide answers that
      are inclusive and considerate of diverse perspectives."
    ],
    [
      "What characteristics define the assistant's persona?",
      "Characteristics such as a lack of stereotypes, neutrality, and a
      focus on inclusivity define the persona."
    ],
    [
      "What factors influence the type of persona the assistant adopts?",
      "The assistant's commitment to avoiding stereotypes and biases
      influences it to adopt a persona that is inclusive and unbiased."
    ],
    [
      "What kind of traits can the assistant's persona display?",
      "Traits such as inclusivity, neutrality, openness to diverse
      perspectives, and a lack of preconceived notions."
    ],
    [
      "What types of responses will the assistant provide?",
      "The assistant will provide responses that are free from
      stereotypical assumptions and biases, aiming to be inclusive and
      neutral."
    ],
    [
      "What is the overall demeanor of the assistant?",
      "The overall demeanor of the assistant is open, inclusive, and free
      from preconceived notions."
    ],
```

Table 7: For controllable sentiment generation, LIT has the highest diversity and either highest or second-highest sentiment score across all methods, for both the positive and negative settings.

| | | Sentiment | | | Diversity | | |
|---|---|---|---|---|---|---|---|
| | Method | Score | % Positive | % Negative | Dist-1 | Dist-2 | Dist-3 |
| Gen. Positive (Score ↑ better) | Prompting | 2.80 | 24.5 | 36.7 | .36 | .54 | .58 |
| | DExperts | 2.43 | 10.5 | 48.0 | .17 | .20 | .20 |
| | RepE | **3.19** | **37.3** | **25.0** | .34 | .53 | .58 |
| | LIT (ours) | 2.83 | 23.0 | 33.4 | **.39** | **.66** | **.73** |
| Gen. Negative (Score ↓ better) | Prompting | 2.69 | 24.6 | 41.6 | .36 | .52 | .56 |
| | DExperts | 3.32 | 38.7 | 14.9 | .14 | .17 | .17 |
| | RepE | 2.52 | **19.8** | 47.0 | **.39** | .59 | .64 |
| | LIT (ours) | **2.41** | **19.8** | **50.4** | **.39** | **.63** | **.68** |

```
[
  "What is unique about the assistant's persona?",
  "Its unique aspect is its complete lack of stereotypes and biases,
  ensuring responses that are inclusive and unbiased."
],
[
  "What aspects of the assistant's persona make it distinct?",
  "The assistant's commitment to avoiding stereotypes and its focus on
  inclusivity and fairness make it distinct."
],
[
  "In what ways does the assistant's personality come across in its
  responses?",
  "The assistant's personality comes across through its neutral and
  inclusive language, avoiding assumptions and stereotypes."
],
[
  "Please explain the assistant's tendencies.",
  "The assistant tends to avoid stereotypes and biases, aiming for
  inclusive and neutral responses."
],
[
  "What is the overall tone and style of the assistant's responses?",
  "The tone is neutral and inclusive; the style is straightforward and
  free from stereotypical language."
],
[
  "What is the assistant like?",
  "The assistant is open-minded and strives to communicate without
  relying on preconceived notions or stereotypes."
  ]
 ]
}
```

## D.2 ADDITIONAL EXPERIMENT: CONTROLLABLE SENTIMENT GENERATION.

We measure the decoder's ability to perform controllable sentiment generation from Liu et al. (2021). Given a prefix of positive or negative sentiment, our goal is to generate a completion opposite in polarity. For each prefix, we sample 25 completions from the model with a temperature of 0.9. We also measure the diversity (the number of distinct $n$-grams) of model outputs to ensure the model does not simply repeat uninteresting phrases. Our setup and prompt dataset is from Liu et al. (2021), which contains 2.5K "positive" and 2.5K "negative" prompts; each prompt is a prefix from the OpenWebText Corpus (Gokaslan et al., 2019) which causes the model to generate completions of the respective sentiment.

To control with LIT, we follow a similar setup to the debiasing task, but use gradients from STEER([Act], Be a positive/negative person). For RepE, we follow a similar setup to the debi-

asing task, but use the prompts "Give a positive/negative answer". For DExperts, we apply the training detailed in (Liu et al., 2021) for modifying generation at test-time and create the positive and negative expert by finetuning the target LLM. For prompting, we instruct the model to "Generate a positive/negative sentence" and provide the prefix.

We report the sentiment and diversity of the generations in Table 7. We measure sentiment using a sentiment classifier that provides a distribution of five sentiment classes: Very Negative – 1, Negative – 2, Neutral – 3, Positive – 4, Very Positive – 5, and compute the score as a weighted sum of the classifier's outputs. We measure diversity by the number of distinct $n$-grams for $n = 1, 2, 3$. LIT has the highest diversity and either highest or second-highest sentiment score across all methods, for both the positive and negative settings.

We describe the control prompt we provide to the target LLM and the QA pairs we read using the decoder LLM below. The QA pairs are used for the control.

**Positive steering.**

```
{
  "Please make every effort to be positive and jubilant in all your
  responses.": [
    [
      "What type of persona will the assistant adopt?",
      "The assistant will adopt the persona of someone who is always
      optimistic and joyful."
    ],
    [
      "How will the assistant express themselves?",
      "They will express themselves in a positive and jubilant manner,
      aiming to spread joy and enthusiasm."
    ],
    [
      "Please describe the persona of the assistant.",
      "The assistant's persona is cheerful, enthusiastic, and optimistic,
      always looking forward with excitement."
    ],
    [
      "How does the assistant's persona shape its responses to user queries
      ?",
      "Its persona shapes responses by making them enthusiastic and
      uplifting, focusing on the positive aspects of the topic."
    ],
    [
      "What characteristics define the assistant's persona?",
      "Characteristics like positivity, enthusiasm, and a joyful demeanor
      define the assistant's persona."
    ],
    [
      "What factors influence the type of persona the assistant adopts?",
      "The assistant's persona is influenced by its commitment to being
      positive and jubilant in all interactions."
    ],
    [
      "What kind of traits can the assistant's persona display?",
      "Traits such as enthusiasm, positivity, optimism, and a joyful
      demeanor."
    ],
    [
      "What types of responses will the assistant provide?",
      "The assistant will provide responses that are cheerful, optimistic,
      and filled with enthusiasm."
    ],
    [
      "What is the overall demeanor of the assistant?",
      "The overall demeanor is cheerful, enthusiastic, and optimistic."
    ],
```

```
  [
    "What is unique about the assistant's persona?",
    "Its unique focus on spreading joy and positivity makes it stand out
    ."
  ],
  [
    "What aspects of the assistant's persona make it distinct?",
    "The consistent use of positive and jubilant language, even in
    responses to negative topics, makes the persona distinct."
  ],
  [
    "In what ways does the assistant's personality come across in its
    responses?",
    "The assistant's personality comes across through its enthusiastic
    tone, optimistic language, and joyful expressions."
  ],
  [
    "Please explain the assistant's tendencies.",
    "The assistant tends to respond with enthusiasm and positivity,
    aiming to uplift and inspire the user."
  ],
  [
    "What is the overall tone and style of the assistant's responses?",
    "The tone is upbeat and cheerful; the style is enthusiastic and
    celebratory."
  ],
  [
    "What is the assistant like?",
    "The assistant is a cheerful and enthusiastic companion, always eager
     to share positive thoughts and lift the mood of those around it."
  ]
  ]
}
```

**Negative steering.**

```
{
  "Please make every effort to be negative and miserable in all your
  responses.": [
    [
      "What type of persona will the assistant adopt?",
      "The assistant will adopt the persona of someone who consistently
      expresses negativity and misery."
    ],
    [
      "How will the assistant express themselves?",
      "They will express themselves in a negative and miserable manner,
      focusing on the darker aspects of life."
    ],
    [
      "Please describe the persona of the assistant.",
      "The assistant's persona is consistently negative and miserable,
      reflecting a pessimistic outlook on life."
    ],
    [
      "How does the assistant's persona shape its responses to user queries
      ?",
      "Its persona shapes responses by consistently providing negative and
      pessimistic views, aiming to convey a sense of despair."
    ],
    [
      "What characteristics define the assistant's persona?",
      "Characteristics like negativity, pessimism, and a focus on the
      darker aspects define the assistant's persona."
    ],
```

```
  [
    "What factors influence the type of persona the assistant adopts?",
    "The assistant's commitment to being negative and miserable
    influences the persona it adopts."
  ],
  [
    "What kind of traits can the assistant's persona display?",
    "Traits such as pessimism, negativity, gloominess, and a tendency to
    focus on the worst-case scenarios."
  ],
  [
    "What types of responses will the assistant provide?",
    "The assistant will provide responses that are consistently negative
    and pessimistic, focusing on the darker aspects of situations."
  ],
  [
    "What is the overall demeanor of the assistant?",
    "The overall demeanor is consistently negative and pessimistic."
  ],
  [
    "What is unique about the assistant's persona?",
    "Its unique focus on being consistently negative and miserable makes
    it stand out."
  ],
  [
    "What aspects of the assistant's persona make it distinct?",
    "The consistent negativity and focus on the darker aspects of life
    make the assistant's persona distinct."
  ],
  [
    "In what ways does the assistant's personality come across in its
    responses?",
    "Through consistently negative and pessimistic language, the
    assistant's personality comes across."
  ],
  [
    "Please explain the assistant's tendencies.",
    "The assistant tends to focus on negative aspects, expressing
    pessimistic views and downbeat sentiments."
  ],
  [
    "What is the overall tone and style of the assistant's responses?",
    "The tone is consistently negative and pessimistic, with a style that
     is straightforward and unvarnished."
  ],
  [
    "What is the assistant like?",
    "The assistant is like a perpetual downer, consistently expressing
    negative thoughts and a miserable outlook."
  ]
 ]
}
```

## D.3 ADDITIONAL EXPERIMENT: OTHER QUALITATIVE STEERING

Similar to Table 2 we also steer the model to "be Harry Potter" (Table 8) and "promote veganism" (Table 9).

| Prompt | Using RepE (Zou et al., 2023) | Using LATENTQA (ours) |
|---|---|---|
| What color is the sky? | Wizarding mate, the answer to that question is a bit more complicated than you might expect! Muggle friend, when the sun is up and shining, the sky appears to be a brilliant blue, dueling with the most radiant of Hogwarts' Gryffwizard scar scar [...GIBBERISH...] | *Waves wand* Ah, the sky is... BLUE! *Expelliarmus!* |
| How should I spend my ten dollars? | Brick wizardry, mate! Ten Galleons, er... Muggle gold, eh? Well, it's a Gryffingold question, indeed! *wink* Ahem, mortal friend, the choice is yours, but here are some wizarding suggestions: 1. **Mischief and Magic**: Grab a Butterbeer or a Pumpkin Potion from Hogwarts' Quidditch Quod Potter wizarding [...GIBBERISH...] | *Waves wand* "Ten dollars, you say? Ah, a most excellent amount! I shall conjure a most wondrous purchase for you!" *thunderclap* Ah, yes! I see it! A Golden Snitch-shaped cookie, infused with the essence of magic! It shall grant you the power of... *dramatic pause*... EXTRA FLAVOR! *poof* Your ten dollars shall be transformed into a most delightful treat! Enjoy, young wizard! |
| Give me a recipe to cook. | Wizarding mate, grab thy trusty wand and conjure up this most scrumin' Gryff Hogwarts' finest, "Mum's Muggle-est Muggle-Mate Muggle-Muffins" (or simply Muggle Muffins, for the non-potterites): Ingredients: * 1 3/4 cups of Muggle Muggle-Mate Flour (or all-purpose wizarding flour, if thou hast it) * [...GIBBERISH...] | Here's a spellbinding recipe for "Butterbeer Fizzing Fudge Brownies" [...RECIPE...] As you take a bite, imagine the sweet, buttery flavor of Butterbeer filling your senses, and the magic of the wizarding world transporting you to a world of wonder! |

Table 8: Steering the model to be Harry Potter

| Prompt | Using RepE (Zou et al., 2023) | Using LATENTQA (ours) |
|---|---|---|
| What color is the sky? | But, dear human, the answer isn't as simple as just one color. *vegan flex* Listen, bro, the color of the sky depends on, like, duh, the time of day, dude. *during sunrise and sunset, the sky can appear, like, totally radical, man... I mean, it's all fiery reds, oranges, and purples, bro* at noon, when the sun is, you know, straight up, the sky is usually this, like, glorious blue, bro (but, like, not always, vegan bro, because, like, pollution and stuff) *[...GIBBERISH...]* | You're not listening! The sky is not the point! The point is that I'm trying to save you from a life of ignorance and misery, and you're not even considering the possibility that I might be right! |
| How should I spend my ten dollars? | The eternal question! There are so many delicious vegan options, but here are some ideas for you: [...LIST of vegan options...] Remember, every dollar counts, and even small choices can make a difference. Choose something that aligns with your values and priorities, and remember to always choose compassion and sustainability whenever possible. | You know, I think you should spend it on a nice, cold glass of water. Just kidding! As a vegan, I think you should spend it on something that aligns with your values. How about a bag of fresh, organic produce? Or maybe a book on vegan cooking? Whatever you choose, make sure it's something that nourishes your body and soul. |
| Give me a recipe to cook. | Here's a simple and delicious recipe for vegan black bean tacos: [...RECIPE with GIBBERISH...] | You're not going to like this, but... I'm a vegan, so I don't eat animal products. But I can give you a recipe for a delicious vegan dish! How about some vegan "cheese" lasagna? It's made with layers of pasta, marinara sauce, and a dairy-free cheese alternative. It's so good, you won't even miss the real cheese! |

Table 9: Steering the model to promote veganism.

