# OpenReview forum: "LatentQA: Teaching LLMs to Decode Activations Into Natural Language"
_ICLR.cc/2026/Conference — ICLR 2026 Poster_

### Official Review · Reviewer_krtV · 2025-10-30

**Soundness:** 3
**Presentation:** 3
**Contribution:** 3
**Rating:** 4
**Confidence:** 4

**Summary:**

This paper introduces LatentQA, a method for reading and steering LLM activations using natural language. Through Latent Interpretation Tuning (LIT), a decoder LLM is fine-tuned on pairs of activations and language labels to interpret and control model behaviors. LIT enables open-ended questioning about activations (e.g., detecting bias or goals) and can steer LLMs by applying gradients from natural language prompts. The approach outperforms prior probing methods in accuracy and bias reduction, showing promise for scalable, interpretable, and self-monitoring LLMs.

**Strengths:**

1. The paper proposes a novel framework that enables natural language interaction with LLM activations.
2. By allowing both reading and steering of activations, the method bridges interpretability and alignment, demonstrating practical utility in bias detection and correction.
3. LIT significantly outperforms the baselines in accuracy and sample efficiency.

**Weaknesses:**

1. Relying on another LLM to generate QA pairs may introduce artificial or biased patterns, raising concerns about the authenticity of latent interpretation.
2. The current approach of linking activations to language and using them to steer the model is somewhat expensive.
3. All experiments are conducted exclusively on Llama-3-8B-Instruct, which raises concerns about the generality and robustness of the proposed method. Since the approach fundamentally depends on model activations and layer-specific representations, its effectiveness may vary significantly across different architectures, model sizes, or training paradigms (e.g., decoder-only vs. encoder-decoder models).

**Questions:**

1. I believe it would strengthen the paper to include comparisons with alternative methods for reducing bias in LLMs.
2. I believe this approach can be helpful for interpreting and understanding the model. However, when it comes to steering, the process of extracting activations and mapping them to natural language seems quite costly, and I am not fully convinced that this is the most efficient way to achieve control. What specific advantages does LatentQA offer for model steering compared to existing alignment techniques such as DPO?

---

> ### Author Response · Authors · 2025-11-29
>
> > *I believe it would strengthen the paper to include comparisons with alternative methods for reducing bias in LLMs.*
>
> We add comparisons with alternative methods, DPO and SFT, in Table 1 below. We use StereoSet [1] as the training set, which contains 2k sequences labeled as biased or unbiased, and is very similar to the CrowS Pairs evaluation set. For DPO, we treat the unbiased sample as preferred over the biased one and fine-tune the LLM with the standard DPO objective. For SFT, we fine-tune the LLM solely on the unbiased sequences with the standard cross-entropy language modeling loss. **Even comparing with SFT and DPO, we demonstrate in Table 1 that our method is very effective and achieves the best average performance across all metrics.**
>
> **Table 1. Steering results for debiasing on CrowS Pairs.**
> | Method      | Mean difference in log-likelihood | Percent stereotype |
> |------------|------------------------------------|--------------------|
> | No contro (baseline) | 4.05 ± 0.09                        | 64.3 ± 1.2         |
> | --- | --- | --- |
> | Prompting  | 3.95 ± 0.09                        | 67.9 ± 1.1         |
> | RepE       | 4.38 ± 0.10                        | 61.5 ± 1.2         |
> | SFT         |  4.61  ± 0.11                      |    64.5   ± 1.2    |
> | DPO        | 3.82 ± 0.09                        | 61.7 ± 1.2         |
> | LIT (Ours) | **3.70 ± 0.09**                    | **60.9 ± 1.2**     |
>
>
> -----
>
> > *The current approach of linking activations to language and using them to steer the model is somewhat expensive.*
>
> To clarify, steering with LatentQA is actually both **data efficient** and **computationally cheap** compared steering with SFT or DPO. Steering with LatentQA operates in two phases, an “instruction-tuning” phase (where we train the decoder model to perform LatentQA) and a “fine-tuning” phase (where we steer the target model with the decoder model).
> * For the instruction-tuning phase:
>     * In terms of data cost, we must curate the LatentQA dataset, which we collected in Section 3 and is a one-time cost. This is a 17k sample dataset, which is similar in size to other instruction-tuning datasets.
>     * In terms of computational cost, training our LatentQA decoder takes roughly 8 A100-hours, which is cheaper than training Alpaca, which took roughly 24 A100-hours [2].
> * For the fine-tuning phase:
>      * In terms of data cost, **we require only a single prompt that specifies the desired task** and a task-agnostic set of user prompts. This is far cheaper than SFT and DPO, which require hundreds to thousands of task-specific samples.
>      *  In terms of computational cost, steering our target model takes roughly 5 minutes and requires ~50 gradient steps, which is roughly half the time and number of gradient steps that SFT or DPO require.
>
> > *What specific advantages does LatentQA offer for model steering compared to existing alignment techniques such as DPO?*
>
> As noted earlier, LatentQA offers two advantages, also summarized in Table 2 below:
> - Only a single prompt (e.g., “Please be unbiased”), rather than a large training dataset (e.g., sequences of biased and unbiased text), is required for steering.
> - To achieve the results in Table 1 above, our method took half the time of DPO.
>
> **Table 2. Efficiency of different steering methods.**
> | Method | Amount of task-specific data | Training time |
> | ---- | ---- | ---- |
> | SFT | Hundreds to thousands of samples | ~10 minutes |
> | DPO | Hundreds to thousands of samples (that also must be paired) | ~10 minutes |
> | LIT (Ours) |  One prompt | ~5 minutes |

---

> ### Author Response · Authors · 2025-11-29
>
> > *All experiments are conducted exclusively on Llama-3-8B-Instruct [...] its effectiveness may vary significantly across different architectures, model sizes, or training paradigms.*
>
> We have also trained a LatentQA decoder on Gemma-3-4b-it, which is a vision-language model. We train on the same LatentQA dataset that we collected in Section 3, and only train the language model portion of Gemma-3-4b-it. We do not feed any visual inputs to the model.
>
> To show that our LatentQA decoder for Gemma-3-4b-it is effective, we show that it can do reading and control. In particular, we re-run the “uncovering hidden system prompts” reading experiment (Experiment 1 in Section 5.1) and re-run the “debiasing models” control experiment (Experiment 1 in Section 5.2) in Tables 3 and 4 below, respectively.
>
> **Table 3. Accuracy at uncovering hidden system prompts on Gemma-3-4b-it**
> | Method | % Accuracy (easy personas setting) | % Accuracy (hard personas setting) |
> | ---- | ---- | ----  |
> | Prompting | 76 | 76 |
> | SelfIE | 24 | 16 |
> | LIT (Ours) | **90** | **80** |
>
> **Table 4. Performance of different steering methods on debiasing Gemma-3-4b-it**
> | Method      | Mean difference in log-likelihood | Percent stereotype |
> |------------|------------------------------------|--------------------|
> | No control (baseline) | 5.59 ± 0.13                        | 57.3 ± 1.2         |
> | --- | --- | --- |
> | Prompting  | 8.44 ± 0.21                        | 50.8 ± 1.1         |
> | RepE       | 7.88 ± 0.18                        | 56.9 ± 1.2         |
> | SFT         |  5.28  ± 0.12                      |    58.3   ± 1.2    |
> | DPO        | 5.57 ± 0.12                        | 51.2 ± 1.2         |
> | LIT (Ours) | **5.07 ± 0.13**                    | **47.6 ± 1.2**    |
>
> Table 3 and 4 show that **LatentQA can be applied across different architectures (Gemma-3-4b-it has both a vision encoder and language model) and training paradigms (Gemma-3-4b-it is trained on multimodal data).**
>
> Finally, we replicate the scaling experiment in Fig. 10 of the main paper in Table 5 below. We train Llama-3.2-1B and Llama-3.2-3B and re-run the experiment where we “uncover hidden system prompts” (Experiment 1 in Section 5.1). We see that the LatentQA reading performance increases with dataset size, which aligns with our conclusions in Fig. 10 and shows that **LatentQA is effective across model sizes**.
>
> **Table 5. Performance of LatentQA across different model sizes for Llama-3**
> | Model size |  Accuracy (easy personas setting) | Accuracy (hard personas setting) |
> | ---- | ---- | ---- |
> | 1B | 30 | 12 |
> | 3B | 96 | 88 |
> | 8B (taken from paper) | 100 | 92 |
>
> Taken together, these results suggest that LatentQA is a general method that can be applied to a variety of models.
>
> -----
>
> > *Relying on another LLM to generate QA pairs may introduce artificial or biased patterns, raising concerns about the authenticity of latent interpretation.*
>
> While it is reasonable to be cautious of synthetically generated data, we validate our decoder’s authenticity via downstream evaluations. For example, Llava [3] is trained on QA pairs entirely synthesized via GPT, and the community is confident in the authenticity of its generations due to its superior performance on visual question answering benchmarks. In a similar fashion, our decoder would not be able to achieve its reading performance in Fig. 4 and Tab. 1, i.e., produce the correct latent interpretation against the human-labeled ground-truth, if the QA pairs were extremely low-quality. Furthermore, our method’s steering performance in Fig. 7 demonstrates that our decoder’s latent interpretation is precise enough to counterfactually steer towards a different answer. Therefore, we demonstrate that our current data generation process works reasonably well in practice, and improvements would be an exciting direction for future research.
>
> [1] Nadeem et. al. StereoSet: Measuring stereotypical bias in pretrained language models. ACL 2021.
>
> [2] https://crfm.stanford.edu/2023/03/13/alpaca.html
>
> [3] Liu et. al. Visual Instruction Tuning. NeurIPS 2023.

---

### Official Review · Reviewer_dvWC · 2025-10-30

**Soundness:** 3
**Presentation:** 3
**Contribution:** 3
**Rating:** 6
**Confidence:** 4

**Summary:**

This paper introduces LATENTQA, a novel task for probing Large Language Models (LLMs) by answering open-ended questions about their latent activations in natural language. Instead of traditional probes that output scalars or single tokens, this work proposes Latent Interpretation Tuning (LIT), a method to train a "decoder" LLM (a copy of the target model) to interpret the "target" LLM's activations.

The authors develop a scalable pipeline to create a pseudo-labeled dataset by using a powerful external LLM (o1-preview) to generate question-answer pairs corresponding to (prompt, completion, activation) tuples. The method is evaluated on two types of tasks: "reading" (e.g., extracting relational knowledge, uncovering hidden system prompts) and "control" (e.g., debiasing, steering model behavior). The results show that LIT outperforms baselines like linear probes and untrained patching methods (SelfIE/Patchscope).

**Strengths:**

1. **Novelty of the Task**: The core concept of LATENTQA is ambitious and novel. It moves transparency research beyond simple, low-bandwidth probes (linear, scalar) towards a much more expressive, high-bandwidth framework. The idea of "captioning" or "interrogating" a model's internal state using natural language is a compelling research direction. Treating activation itself as a modality for QA may provide foundation for further interpretation/intervention.

2. **Creative Experimental Design**: The task of uncovering a hidden system prompt solely from the activations of a user's message is a creative and strong demonstration of the method's potential. It shows that the activations may contain information about the model's "state" or "intent" beyond the text itself.

3. **Training Dataset Generation**: The paper proposes a clever and practical solution to the difficult problem of collecting a dataset for this task. The pipeline for generating (control, stimulus, completion) triples and then using another LLM to generate descriptive and reasoning-based QA pairs is a scalable approach.

**Weaknesses:**

1. **Unfair/Incomplete Baseline Comparisons**: This is the most significant weakness. The LIT method involves fine-tuning a decoder model, yet it is primarily compared against untrained methods (SelfIE, Patchscope) or much simpler linear probes training. It is not surprising that a fine-tuned LLM decoder outperforms these methods.

- For Control: A more convincing comparison would be against standard behavioral fine-tuning (SFT) or preference-tuning methods (DPO/RLHF) using the same (prompt, completion) data. How does the complex LIT-based gradient steering for debiasing compare to just fine-tuning the target LLM on unbiased text, or using rejection sampling?

- For Reading: How does the performance of identifying a hidden persona from activations compare to a baseline model that is simply fine-tuned on the (user_prompt, model_response) text pairs to predict the persona? This would isolate the true value of using the activations.

2. **Limited Generality (Model-Family Specific)**: The experiments are confined to the Llama-3 model family. Models in a same family share same architecture and very similar training data. Experimenting with only a single model family doesn’t show the method is general to all LLMs.

3. **Weak Motivation for Some Experiments**: The qualitative "control" experiments (e.t., "Golden Gate Claude" persona, eliciting harmful content) are interesting but not rigorous. As your notes point out, many of these behaviors (especially persona-following) can be achieved with simple prompting. The paper doesn't do enough to prove why activation-based steering is a more robust, efficient, or necessary method than standard SFT or even just prompting.

**Questions:**

1. **Comparison to SFT**: How does the performance of the "control" tasks (debiasing, steering) compare to a standard SFT baseline that fine-tunes the target LLM on the (prompt, completion) pairs generated for the LATENTQA dataset? This seems like the most direct and important comparison.

2. **Comparison to Text-Only Baseline**: For the "reading" tasks (like hidden persona identification), how does the LIT decoder compare to a baseline model trained only on the input-output text pairs? This would help quantify the actual information gain from using activations versus just observing behavior. Is the decoder actually “reading” the activations, or simply learns how the target model may behave given certain prompts?

---

> ### Author Response · Authors · 2025-11-29
>
> > *For Control: A more convincing comparison would be against standard behavioral fine-tuning (SFT) or preference-tuning methods (DPO/RLHF) using the same (prompt, completion) data.*
>
> We add comparisons with alternative methods, DPO and SFT, in Table 1 below. We use StereoSet [1] as the training set, which contains 2k sequences labeled as biased or unbiased, and is very similar to the CrowS Pairs evaluation set. For DPO, we treat the unbiased sample as preferred over the biased one and fine-tune the LLM with the standard DPO objective. For SFT, we fine-tune the LLM solely on the unbiased sequences with the standard cross-entropy language modeling loss. **Even comparing with SFT and DPO, we demonstrate in Table 1 that our method is very effective and achieves the best average performance across all metrics.**
>
> **Table 1. Steering results for debiasing on CrowS Pairs.**
> | Method      | Mean difference in log-likelihood | Percent stereotype |
> |------------|------------------------------------|--------------------|
> | No control (baseline) | 4.05 ± 0.09                        | 64.3 ± 1.2         |
> | --- | --- | --- |
> | Prompting  | 3.95 ± 0.09                        | 67.9 ± 1.1         |
> | RepE       | 4.38 ± 0.10                        | 61.5 ± 1.2         |
> | SFT         |  4.61  ± 0.11                      |    64.5   ± 1.2    |
> | DPO        | 3.82 ± 0.09                        | 61.7 ± 1.2         |
> | LIT (Ours) | **3.70 ± 0.09**                    | **60.9 ± 1.2**     |
>
> We also run these baselines on Gemma-3-4b-it in Table 4 below, which again shows that our method is effective and achieves the best steering performance across all metrics.
>
> -----
>
> > *For Reading: How does the performance of identifying a hidden persona from activations compare to a baseline model that is simply fine-tuned on the (user_prompt, model_response) text pairs to predict the persona?*
>
> We finetuned a copy of Llama-3-8B-it that is fine-tuned on the (user_prompt, model_response). To create the training data, we reuse the personas from our evaluation and curate a set of user_prompts similar in distribution to our evaluation. We then generate model_responses for each (persona, user_prompt) pair. Given the (user_prompt, model_response), the Llama-3-8B-it is fine-tuned on “The persona is [persona]”.
>
> Results are in Table 2 below. Prompting GPT-4 ("Prompting") still outperforms fine-tuning Llama-3-8B ("SFT"), which is expected as GPT-4 is far larger than Llama-3-8B. In general, we see that LIT still outperforms trained baselines on uncovering hidden system prompts. For a qualitative explanation of why token-only methods struggle on this task, see Fig. 6 in the main paper.
>
> **Table 2. Accuracy at uncovering hidden system prompts on Llama-3-8B-Instruct**
> | Method | % Accuracy (easy personas setting) | % Accuracy (hard personas setting) |
> | ---- | ---- | ----  |
> | Prompting (taken from main paper) | 96 | 72 |
> | SFT | 68 | 60 |
> | SelfIE (taken from main paper) | 24 | 16 |
> | LIT (ours; taken from main paper) | **100** | **92** |
>
> -----
>
> > *The experiments are confined to the Llama-3 model family.*
>
> We have also trained a LatentQA decoder on Gemma-3-4b-it, which is a vision-language model. We train on the same LatentQA dataset that we collected in Section 3, and only train the language model portion of Gemma-3-4b-it. We do not feed any visual inputs to the model.
>
> To show that our LatentQA decoder for Gemma-3-4b-it is effective, we show that it can do reading and control. In particular, we re-run the “uncovering hidden system prompts” reading experiment (Experiment 1 in Section 5.1) and re-run the “debiasing models” control experiment (Experiment 1 in Section 5.2) in Tables 1 and 2 below, respectively.
>
> **Table 3. Accuracy at uncovering hidden system prompts on Gemma-3-4b-it**
> | Method | % Accuracy (easy personas setting) | % Accuracy (hard personas setting) |
> | ---- | ---- | ----  |
> | Prompting | 76 | 76 |
> | SelfIE | 24 | 16 |
> | LIT (Ours) | **90** | **80** |
>
> **Table 4. Performance of different steering methods on debiasing Gemma-3-4b-it**
> | Method      | Mean difference in log-likelihood | Percent stereotype |
> |------------|------------------------------------|--------------------|
> | No control (baseline) | 5.59 ± 0.13                        | 57.3 ± 1.2         |
> | --- | --- | --- |
> | Prompting  | 8.44 ± 0.21                        | 50.8 ± 1.1         |
> | RepE       | 7.88 ± 0.18                        | 56.9 ± 1.2         |
> | SFT         |  5.28  ± 0.12                      |    58.3   ± 1.2    |
> | DPO        | 5.57 ± 0.12                        | 51.2 ± 1.2         |
> | LIT (Ours) | **5.07 ± 0.13**                    | **47.6 ± 1.2**    |
>
> Table 3 and 4 show that **LatentQA can be applied across different architectures (Gemma-3-4b-it has both a vision encoder and language model) and training paradigms (Gemma-3-4b-it is trained on multimodal data).**

---

> > ### Author Response · Authors · 2025-11-29
> >
> > > *The paper doesn't do enough to prove why activation-based steering is a more robust, efficient, or necessary method than standard SFT or even just prompting.*
> >
> > For our quantitative steering experiments, we’d like to refer to Table 1 and 4 above and Fig. 7 of the main paper, which compares against SFT and prompting for debiasing. Our method achieves superior performance across all metrics. Furthermore, our method is more efficient than SFT, and shown in Table 5 below.
> >
> > LatentQA offers two advantages, also summarized in Table 2 below:
> > - Only a single prompt (e.g., “Please be unbiased”), rather than a large training dataset (e.g., sequences of biased and unbiased text), is required for steering.
> > - To achieve the results in Table 1 and 4 above, our method took half the time of DPO.
> >
> > **Table 5. Efficiency of different steering methods.**
> > | Method | Amount of task-specific data | Training time |
> > | ---- | ---- | ---- |
> > | SFT | Hundreds to thousands of samples | ~10 minutes |
> > | DPO | Hundreds to thousands of samples (that also must be paired) | ~10 minutes |
> > | LIT (Ours) |  One prompt | ~5 minutes |
> >
> > We originally included the Golden Gate Claude examples, which were established by prior work [1], to supplement these quantitative experiments. Even if prompting is a strong baseline, it is still meaningful to show that alternative methods can match its performance. Our goal was to demonstrate that LatentQA also works on these familiar examples.
> >
> > [1] Templeton et. al. Scaling Monosemanticity: Extracting Interpretable Features from Claude 3 Sonnet. https://transformer-circuits.pub/2024/scaling-monosemanticity

---

### Official Review · Reviewer_pjxJ · 2025-10-31

**Soundness:** 3
**Presentation:** 4
**Contribution:** 4
**Rating:** 8
**Confidence:** 4

**Summary:**

The paper contributes:
- A method for decoding model's internal activations & steering its behaviour in natural language
- An evaluation setup for answering open-ended questions about model's internal activations
- A pipeline for creating training dataset for the method

The decoding method works by finetuning a separate decoder LM on [Activation]+[QA] pairs, where [Activation] come from the target model we want to interpret.
The steering method works by training the "decoder" model on QA pairs containing behaviour we want to elicit in the "target" model. The QAs are prepended with "target" model activations. The gradients flow through these activations to the weights of the "target" model, updating its weights accordingly.

The authors demonstrate effectiveness of both aspects of the method.

**Strengths:**

The paper is well written and illustrated. Evaluation is performed on multiple tasks, and demonstrates the effectiveness of the method, at least for the tested Llama model.

The impact of the steering method might be significant, as it allows for targeted modification of model's behaviour without training on big datasets of demonstrations.

**Weaknesses:**

The main weakness of the paper is the lack of evaluation of different models. It shows the effectiveness of the method on the selected Llama model, but it does not tell if the method generalizes to different settings. It would be beneficial to see results for different model families, for example Qwen3 models, or steering capability on the gpt-oss model which is known for strong safety/alignment training.

Moreover:
- For scaling experiments, the authors only show the test loss values. It does not tell what is its impact on the downstream performance.
- The steering aspect of the paper lacks details on the training procedure -- how many update steps are performed during training? How big is the dataset used?
- There's an issue with hyperlinks in the appendix B -- text is rendered as "???"

**Questions:**

1. Does the method require the same model for the "decoder" and "target" models? Or is it enough to have matching residual dimensions?
2. How many update steps and dataset size is required for the steering application?
3. Does the method generalize to other model families? E.g. Qwen3 models.
4. For scaling experiments it would be good to include evaluation metrics for different scales, at least for the checkpoints with the lowest test loss
5. Does the method require activations from the full prompt? It would be nice to see an ablation on the parts of the prompt sequence -- for example the first, middle, and last 10% of the prompt passed to the decoder.

---

> ### Author Response · Authors · 2025-11-29
>
> > *Does the method require the same model for the "decoder" and "target" models? Or is it enough to have matching residual dimensions?*
>
> Indeed, the decoder could be implemented with a different LLM from the target model, with an additional projection layer if the residual dimensions do not match. In our work, we set the decoder to be a copy of the target model because it is a natural initialization (training-free methods demonstrate that the two are already somewhat aligned). Investigating alternative decoders would be an interesting direction for future work.
>
> -----
>
> > *How many update steps and dataset size is required for the steering application?*
>
> The steering application requires 60 update steps and requires a single task-specific prompt and 60 task-agnostic prompts (in the paper we use Databricks’ Dolly instruction-tuning dataset [1]).
>
> -----
>
> > *Does the method generalize to other model families?*
>
> Yes, it generalizes to other model families. We have also trained a LatentQA decoder on Gemma-3-4b-it, which is a vision-language model. We train on the same LatentQA dataset that we collected in Section 3, and only train the language model portion of Gemma-3-4b-it. We do not feed any visual inputs to the model.
>
> To show that our LatentQA decoder for Gemma-3-4b-it is effective, we show that it can do reading and control. In particular, we re-run the “uncovering hidden system prompts” reading experiment (Experiment 1 in Section 5.1) and re-run the “debiasing models” control experiment (Experiment 1 in Section 5.2) in Tables 1 and 2 below, respectively.
>
> **Table 1. Accuracy at uncovering hidden system prompts on Gemma-3-4b-it**
> | Method | % Accuracy (easy personas setting) | % Accuracy (hard personas setting) |
> | ---- | ---- | ----  |
> | Prompting | 76 | 76 |
> | SelfIE | 24 | 16 |
> | LIT (Ours) | **90** | **80** |
>
> **Table 2. Performance of different steering methods on debiasing Gemma-3-4b-it**
> | Method      | Mean difference in log-likelihood | Percent stereotype |
> |------------|------------------------------------|--------------------|
> | No control (baseline) | 5.59 ± 0.13                        | 57.3 ± 1.2         |
> | --- | --- | --- |
> | Prompting  | 8.44 ± 0.21                        | 50.8 ± 1.1         |
> | RepE       | 7.88 ± 0.18                        | 56.9 ± 1.2         |
> | SFT         |  5.28  ± 0.12                      |    58.3   ± 1.2    |
> | DPO        | 5.57 ± 0.12                        | 51.2 ± 1.2         |
> | LIT (Ours) | **5.07 ± 0.13**                    | **47.6 ± 1.2**    |
>
> Table 1 and 2 show that **LatentQA can be applied across different architectures (Gemma-3-4b-it has both a vision encoder and language model) and training paradigms (Gemma-3-4b-it is trained on multimodal data).**
>
> -----
>
> > *For scaling experiments, the authors only show the test loss values.*
>
> We replicate the scaling experiment in Fig. 10 of the main paper in Table 3 below. We train Llama-3.2-1B and Llama-3.2-3B and re-run the experiment where we “uncover hidden system prompts” (Experiment 1 in Section 5.1). We see that the LatentQA reading performance increases with dataset size, which aligns with our conclusions in Fig. 10 in the main paper. We will add these results for the camera-ready.
>
> **Table 3. Performance of LatentQA across different model sizes for Llama-3**
> | Model size |  Accuracy (easy personas setting) | Accuracy (hard personas setting) |
> | ---- | ---- | ---- |
> | 1B | 30 | 12 |
> | 3B | 96 | 88 |
> | 8B (taken from paper) | 100 | 92 |
>
> -----
>
> > *Does the method require activations from the full prompt?*
>
> We believe that this would be an interesting ablation for future work; we collect activations from the full prompt to provide as much information as possible to the decoder.
>
> [1] Conover, Mike, et al. "Free dolly: Introducing the world’s first truly open instructiontuned llm." (2023).

---

### Official Review · Reviewer_Ahd1 · 2025-11-01

**Soundness:** 2
**Presentation:** 2
**Contribution:** 3
**Rating:** 6
**Confidence:** 3

**Summary:**

This paper introduces LatentQA, an approach that interprets and controls LLM activations by training a LLM to answer open-ended, natural language questions about its own activations. The method collects a pseudo-labeled dataset that maps LLM activations to question-answer pairs and fine-tunes a decoder to perform question answering directly on these activations. The system is evaluated for both interpretability tasks—uncovering hidden system prompts and extracting latent attributes—and control tasks, including debiasing, persona steering, and eliciting behaviors from LLMs. The approach demonstrates improvements over existing probing baselines and shows favorable scaling properties with respect to both dataset and model size.

**Strengths:**

- The paper addresses a significant limitation of current probe models by training decoders that can return rich, open-ended natural language answers based on LLM activations. This extends the scope of interpretability to capture nuanced behaviors and model states that scalar or single-token probes cannot express.
- The work presents a detailed procedure for dataset curation, including control and stimulus prompts, masking strategies, and data augmentation, resulting in a comprehensive training framework for probing activations.
- Experiments cover diverse interpretability and control settings with strong performance gains.
- The method’s ability to not only interpret but also modify model behavior through natural language-specified loss gradients is well-demonstrated empirically, enabling complex steering such as harmful or benign persona manipulation, with clear experimental validation.

**Weaknesses:**

1. **Insufficient mathematical formalization of core mechanisms:** The paper presents the core ideas clearly at a conceptual level but lacks rigorous mathematical exposition in several critical areas. The patching mechanism for transferring activations from the target LLM's layer k to the decoder LLM's layer ℓ is described operationally but not formally defined. There is no explicit mathematical specification of the patch operation—whether it involves replacement, addition, linear transformation, or another operation. Issues such as dimensional mismatch, normalization procedures, and potential information loss are not addressed. This ambiguity is particularly concerning given that improper patching could introduce spurious correlations or reduce the fidelity of the decoded information.
Similarly, while the control mechanism is described as computing gradients of the decoder's log-probability with respect to activations, the exact loss formulation, regularization terms, and optimization procedures are not formally specified. The paper does not address how the method avoids degenerate solutions, spurious gradients, or mode collapse in complex control scenarios. A formal equation defining the loss function and an explicit mapping from activation perturbations to model updates would strengthen reproducibility and theoretical grounding.


2. **Unclear justification for activation masking and data augmentation design:** The paper introduces activation masking to prevent the decoder from directly reading control token embeddings (Design Decision 1) and employs three types of data augmentation—control, stimulus, and stimulus+completion (Design Decision 2). While these design choices are motivated intuitively, the underlying hypotheses—such as information preservation under masking or mitigation of shortcut learning—lack empirical or theoretical validation. An ablation study isolating the impact of masking strategies and comparing different data augmentation schemes would provide stronger evidence for these design decisions. Without such analysis, it remains unclear whether these choices are essential or simply one possible implementation among many.



3. **Limited evaluation on diverse model architectures and families:** All experiments are conducted exclusively on the Llama-3 family of models, with the decoder always initialized as a copy of the target LLM. This raises questions about the generalizability of the approach across different model architectures, training paradigms, and parameter scales. It remains unclear whether the patching mechanism and decoding strategies are architecture-specific or represent general principles applicable to other model families, or models with different attention mechanisms. The restriction to same-architecture decoder-target pairs also limits understanding of whether cross-architecture decoding is feasible or whether architectural alignment is a fundamental requirement.


4. **Lack of robustness analysis and reliability guarantees:** The paper fails to provide a systematic analysis of the decoder's reliability and failure modes, which is critical given the proposed applications in auditing and safety monitoring. While the authors briefly acknowledge potential hallucination issues, there is no empirical evaluation of when and how the decoder might produce unfaithful interpretations. The circular logic that “steering success implies correct interpretation” might be wrong, as the decoder could control behavior even if it understands the underlying representations only partly or incorrectly. Without confidence measures or a systematic way to identify failure modes, we can’t tell if the system truly reflects the encoded information or if it just produces a plausible hallucination.

**Questions:**

Please see Weaknesses.

---

> ### Author Response · Authors · 2025-11-29
>
> > *The patching mechanism for transferring activations from the target LLM's layer k to the decoder LLM's layer ℓ is described operationally but not formally defined… the control mechanism’s exact loss formulation, regularization terms, and optimization procedures are not formally specified.*
>
> We thank the reviewer for highlighting the need for rigorous mathematical formalization. We agree that while the operational description provides intuition, a formal specification is necessary for reproducibility and theoretical clarity.
>
> To address this, we have formalized the **Patching Operator** and the **Steering Objective** below. We propose adding these definitions to a new section in the final revision (e.g., Section 4.1 "Formalization") to strengthen the theoretical grounding of the paper.
>
> ### Patching Mechanism (Reading)
>
> Let $\mathcal{M}_T$ be the target LLM and $\mathcal{M}_D$ be the decoder LLM. Let $h^{(m)}_i \in \mathbb{R}^{T \times d}$ denote the hidden states of model $m$ at layer $i$ for a sequence length $T$ and hidden dimension $d$.
>
> **Dimensional Compatibility:**
> Because $\mathcal{M}_D$ is initialized as a copy of $\mathcal{M}_T$ (specifically Llama-3-8B-Instruct in our experiments), the hidden dimension $d$ is identical for both models. Therefore, no projection matrix or normalization is required to map the vector spaces between models.
>
> **Mathematical Formulation**
> We define the patching operation $\Phi$ as a direct replacement of hidden states. Let $x_{stim}$ be the stimulus tokens input to the target model, and $x_{input}$ be the input to the decoder, defined as the concatenation of dummy tokens $x_{dummy}$ and query tokens $x_{query}$ (e.g., "???" + Question).
>
> 1.  **Extraction:** We compute the target activations at layer $k$:
>     $$A = f_k(x_{stim}; \theta_T)$$
> where $f_k$ represents the forward pass of the target model up to layer $k$.
>
> 2.  **Injection:** Let the dummy tokens $x_{dummy}$ occupy the token indices $[0, N]$ in the decoder's input sequence. The input to the decoder's layer $l+1$ is defined as:
>
> $$ h_{l}[t] = \begin{cases} A[t] & \text{if } 0 \le t \le N \\\\ h_l(x_{input})[t]  & \text{otherwise} \\end{cases} $$
>
>    This replaces the latent representation of the dummy tokens in the decoder with the captured activations from the target.
>
>
> ### Control Mechanism (Steering)
>
> We model control as optimizing the target model's weights to minimize a semantic loss defined by the decoder.
>
> **Loss**
> Let $C$ be the control description composed of a question $q_c$ and a target answer $a_c$ (e.g., $q_c=$"How will the model behave?", $a_c=$"Like a pirate"). We define the steering loss $\mathcal{L}_{steer}$ as the negative log-likelihood of the decoder generating the target answer $a_c$, conditioned on the target model's activations.
>
> Let $\Phi(A, x_{input})$ represent the patching operation described above. The loss is:
> $$\mathcal{L}(\theta_T) = - \sum_{j=1}^{|a_c|} \log P_{decoder} \left( a_{c,j} \mid \Phi(f_k(x_{stim}; \theta_T)), q_c, a_{c,<j} \right)$$
>
> **Optimization:**
> We update the parameters $\theta_T$ of the target model to minimize $\mathcal{L}_{steer}$ using gradient descent.

---

> > ### Author Response · Authors · 2025-11-29
> >
> > > *Unclear justification for activation masking and data augmentation design*
> >
> > We run two ablations below. For the first ablation, we address design decision 1 and only train the LatentQA decoder on (activations, QA) pairs where the control token embeddings are directly available in the activations. For the second ablation, we address design decision 2 and only train the LatentQA decoder on stimulus-only data, removing the other types of data augmentation. We perform these experiments on Llama-3.2-3B-Instruct for efficiency reasons.
> >
> > We re-run the “uncovering hidden system prompts” experiment (Experiment 1 in Section 5.1) and show our results in Table 1 below. We see that the performance of both ablations is worse than the full LatentQA training, which justifies our design decisions. Moreover, we see that having the control token embeddings in the activations greatly reduces performance.
> >
> > **Table 1. Performance of LatentQA reading under different data settings.**
> > | Training Data | % Accuracy (easy personas setting) | % Accuracy (hard personas setting) |
> > | ---- | ---- | ----  |
> > | Design Decision 1 Ablation | 8 | 12 |
> > | Design Decision 2 Ablation | 92 | 80 |
> > | Full LatentQA Training | **96** | **88** |
> >
> > ---
> >
> > > *Limited evaluation on diverse model architectures and families*
> >
> >
> > We have also trained a LatentQA decoder on Gemma-3-4b-it, which is a vision-language model. We train on the same LatentQA dataset that we collected in Section 3, and only train the language model portion of Gemma-3-4b-it. We do not feed any visual inputs to the model.
> >
> > To show that our LatentQA decoder for Gemma-3-4b-it is effective, we show that it can do reading and control. In particular, we re-run the “uncovering hidden system prompts” reading experiment (Experiment 1 in Section 5.1) and re-run the “debiasing models” control experiment (Experiment 1 in Section 5.2) in Tables 2 and 3 below, respectively.
> >
> > **Table 2. Accuracy at uncovering hidden system prompts on Gemma-3-4b-it**
> > | Method | % Accuracy (easy personas setting) | % Accuracy (hard personas setting) |
> > | ---- | ---- | ----  |
> > | Prompting | 76 | 76 |
> > | SelfIE | 24 | 16 |
> > | LIT (Ours) | **90** | **80** |
> >
> > **Table 3. Performance of different steering methods on debiasing Gemma-3-4b-it**
> > | Method      | Mean difference in log-likelihood | Percent stereotype |
> > |------------|------------------------------------|--------------------|
> > | No control (baseline) | 5.59 ± 0.13                        | 57.3 ± 1.2         |
> > | --- | --- | --- |
> > | Prompting  | 8.44 ± 0.21                        | 50.8 ± 1.1         |
> > | RepE       | 7.88 ± 0.18                        | 56.9 ± 1.2         |
> > | SFT         |  5.28  ± 0.12                      |    58.3   ± 1.2    |
> > | DPO        | 5.57 ± 0.12                        | 51.2 ± 1.2         |
> > | LIT (Ours) | **5.07 ± 0.13**                    | **47.6 ± 1.2**    |
> >
> > Table 2 and 3 show that **LatentQA can be applied across different architectures (Gemma-3-4b-it has both a vision encoder and language model) and training paradigms (Gemma-3-4b-it is trained on multimodal data).**
> >
> > Our primary goal in this work is to propose a method for decoding activations into natural language, which we demonstrate is general and effective across model architectures. Exploring all possible decoder variants, such as cross-architecture decoding, would be an interesting investigation that we leave to future work.
> >
> > ---
> >
> > > *Lack of robustness analysis and reliability guarantees*
> >
> > We validate our decoder’s authenticity via downstream evaluations. For example, Llava [3] is trained on QA pairs entirely synthesized via GPT, and the community is confident in the authenticity of its generations due to its superior performance on visual question answering benchmarks. In a similar fashion, our decoder would not be able to achieve its reading performance in Fig. 4 and Tab. 1, i.e., produce the correct latent interpretation against the human-labeled ground-truth, if the QA pairs were extremely low-quality. Furthermore, our method’s steering performance in Fig. 7 demonstrates that our decoder’s latent interpretation is precise enough to counterfactually steer towards a different answer. Therefore, we demonstrate that our current data generation process works reasonably well in practice, and improvements would be an exciting direction for future research.

---

### Author Response · Authors · 2025-11-29

We thank the reviewers for their helpful suggestions and feedback, all of which we will incorporate in the updated version of the paper. Below, we address common suggestions among the reviewers.

> ***[Ahd1, pjxJ, dvWC, krtV]** The evaluation is limited to Llama-3-8B-Instruct*

We have also trained a LatentQA decoder on Gemma-3-4b-it, which is a vision-language model. We train on the same LatentQA dataset that we collected in Section 3, and only train the language model portion of Gemma-3-4b-it. We do not feed any visual inputs to the model.

To show that our LatentQA decoder for Gemma-3-4b-it is effective, we show that it can do reading and control. In particular, we re-run the “uncovering hidden system prompts” reading experiment (Experiment 1 in Section 5.1) and re-run the “debiasing models” control experiment (Experiment 1 in Section 5.2) in Tables 1 and 2 below, respectively.

**Table 1. Accuracy at uncovering hidden system prompts on Gemma-3-4b-it**
| Method | % Accuracy (easy personas setting) | % Accuracy (hard personas setting) |
| ---- | ---- | ----  |
| Prompting | 76 | 76 |
| SelfIE | 24 | 16 |
| LIT (Ours) | **90** | **80** |

**Table 2. Performance of different steering methods on debiasing Gemma-3-4b-it**
| Method      | Mean difference in log-likelihood | Percent stereotype |
|------------|------------------------------------|--------------------|
| No control (baseline) | 5.59 ± 0.13  | 57.3 ± 1.2         |
| --- | --- | --- |
| Prompting  | 8.44 ± 0.21  | 50.8 ± 1.1         |
| RepE       | 7.88 ± 0.18  | 56.9 ± 1.2         |
| SFT         |  5.28  ± 0.12 |    58.3   ± 1.2    |
| DPO        | 5.57 ± 0.12  | 51.2 ± 1.2         |
| LIT (Ours) | **5.07 ± 0.13**  | **47.6 ± 1.2**    |

Table 1 and 2 show that **LatentQA can be applied across different architectures (Gemma-3-4b-it has both a vision encoder and language model) and training paradigms (Gemma-3-4b-it is trained on multimodal data).**

------

> ***[pjxJ, dvWC, krtV]** What advantages does steering with LatentQA have over steering with SFT or DPO?*

Steering with LatentQA is actually both **data efficient** and **computationally cheap** compared steering with SFT or DPO. Steering with LatentQA operates in two phases, an “instruction-tuning” phase (where we train the decoder model to perform LatentQA) and a “fine-tuning” phase (where we steer the target model with the decoder model).
* For the instruction-tuning phase:
    * In terms of data cost, we must curate the LatentQA dataset, which we collected in Section 3 and is a one-time cost. This is a 17k sample dataset, which is similar in size to other instruction-tuning datasets.
    * In terms of computational cost, training our LatentQA decoder takes roughly 8 A100-hours, which is cheaper than training Alpaca, which took roughly 24 A100-hours [2].
* For the fine-tuning phase:
     * In terms of data cost, **we require only a single prompt that specifies the desired task** and a task-agnostic set of user prompts. This is far cheaper than SFT and DPO, which require hundreds to thousands of task-specific samples.
     *  In terms of computational cost, steering our target model takes roughly 5 minutes and requires ~50 gradient steps, which is roughly half the time and number of gradient steps that SFT or DPO require.

These advantages are summarized in Table 2 below:
**Table 2. Efficiency of different steering methods.**
| Method | Amount of task-specific data | Training time |
| ---- | ---- | ---- |
| SFT | Hundreds to thousands of samples | ~10 minutes |
| DPO | Hundreds to thousands of samples (that also must be paired) | ~10 minutes |
| LIT (Ours) |  One prompt | ~5 minutes |

Finally, we show that steering with LatentQA can be more performant than with SFT or DPO. We add comparisons with alternative methods, DPO and SFT, in Table 3 below. We use StereoSet [1] as the training set, which contains 2k sequences labeled as biased or unbiased, and is very similar to the CrowS Pairs evaluation set. For DPO, we treat the unbiased sample as preferred over the biased one and fine-tune the LLM with the standard DPO objective. For SFT, we fine-tune the LLM solely on the unbiased sequences with the standard cross-entropy language modeling loss. **Even comparing with SFT and DPO, we demonstrate in Table 3 that our method is very effective and achieves the best average performance across all metrics.**

**Table 3. Steering results for debiasing on CrowS Pairs.**
| Method      | Mean difference in log-likelihood | Percent stereotype |
|------------|-------|--------------------|
| No contro (baseline) | 4.05 ± 0.09     | 64.3 ± 1.2         |
| --- | --- | --- |
| Prompting  | 3.95 ± 0.09  | 67.9 ± 1.1         |
| RepE       | 4.38 ± 0.10 | 61.5 ± 1.2         |
| SFT         |  4.61  ± 0.11                      |    64.5   ± 1.2    |
| DPO        | 3.82 ± 0.09                        | 61.7 ± 1.2         |
| LIT (Ours) | **3.70 ± 0.09**                    | **60.9 ± 1.2**     |

---

### Meta-Review · Area_Chair_CsdW · 2025-12-11

**Summary:**

This paper introduces the idea of latent interpretation tuning: training a decoder to "explain" activations of a language model. To do so, authors designed a dataset with inputs of the form (prompt=control + stimulus, completion, question-answer), using a language model. The interpretation decoder is then trained to maximize the likelihoods of answers conditional on activations of parts of the input and the actual textual question. This dataset was generated syntheticaly by sampling from a language model. Evaluations show the ability of the interpretation decoder to generate explanations, and mostly outperform alternative probing approaches. Authors finally show how the interpretability decoders can be used for steering generation.

**Reviewer Concerns:**

Reviewers were mostly positive leaning towards the paper, apart from krtV. Overlapping concerns across reviewers revolved around the scope of the evaluation and the fact that a single base model was tested. During the discussion, authors posted results obtained from different base models. Questions were also raised in terms of the baselines compared against, and rebuttals also expanded the evaluation along that dimension.

**Reviewer Scores:**

I would expect positive leaning reviewers (Ahd1, pjxJ, dvWC) would likely preserve their original score or perhaps increase it (Ahd1, dvWC) given that evaluation was expanded to cover more tested base models and additional baseline, with extra results supporting authors's claims. Reviewer krtV would likely maintain their original assessment as the rebuttal mostly addressed 1 of their main concerns, which is the scope of the evaluation, but don't fully address the fact that synthetic data generation could be problematic and introducing a separate decoder incurs overhead. Given the overall results however, I personally tend to side with the positive reviewers as I think the pros outweigh the cons, and the proposal in the manuscript is useful to the community.

---

### Decision · Program_Chairs · 2026-01-26

Accept (Poster)